# Macro-from-Micro Planning for High-Quality and Parallelized Autoregressive Long Video Generation

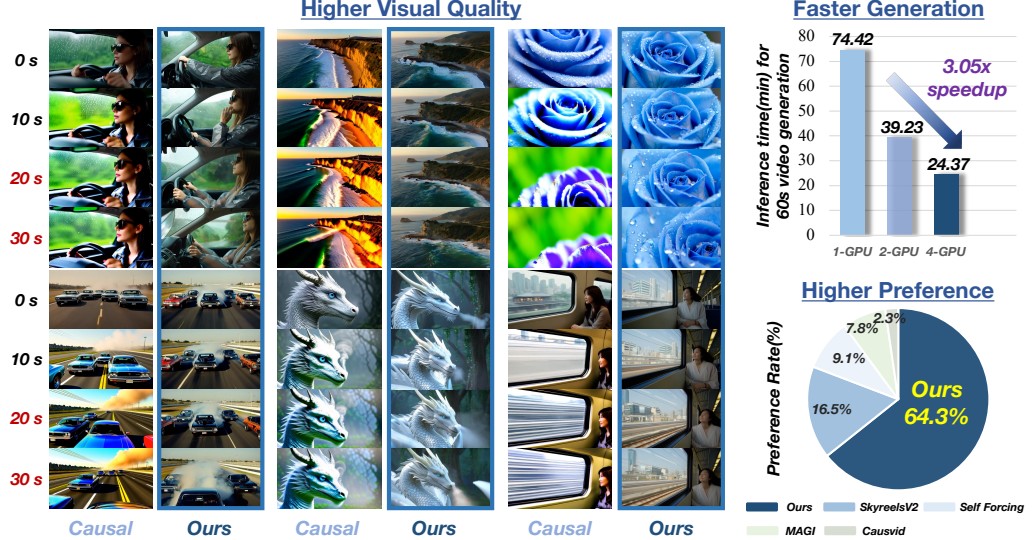

Figure 1: We propose *Macro-from-Micro Planning* (MMPL), a paradigm for long-video generation that achieves higher visual quality, faster speed, and stronger user preference than existing methods. Snapshots at 0s, 10s, 20s, and 30s (left) show robustness against temporal drift—semantic shifts, color changes, and structural artifacts—while quantitative results highlight accelerated multi-GPU inference (top-right) and dominant user preference (bottom-right).

## Abstract

Current autoregressive diffusion models excel at video generation but are generally limited to short temporal durations. Our theoretical analysis indicates that the autoregressive modeling typically suffers from temporal drift caused by error accumulation and hinders parallelization in long video synthesis. To address these limitations, we propose a novel planning-then-populating framework centered on Macro-from-Micro Planning (MMPL) for long video generation. MMPL sketches a global storyline for the entire video through two hierarchical stages: *Micro Planning and Macro Planning*. Specifically, *Micro Planning* predicts a sparse set of future keyframes within each short video segment, offering motion and appearance priors to guide high-quality video segment generation. *Macro Planning* extends the in-segment keyframes planning across the entire video through an autoregressive chain of micro plans, ensuring long-term consistency across video segments. Subsequently, MMPL-based *Content Populating* generates all intermediate frames in parallel across segments, enabling efficient parallelization of autoregressive generation. The parallelization is further optimized by *Adaptive Workload Scheduling* for balanced GPU execution and accelerated autoregressive video generation. Extensive experiments confirm that our method outperforms existing long video generation models in quality and stability. Generated videos and comparison results are in the **Anonymous** Demo page.

## 1 INTRODUCTION

Long video generation is crucial for applications such as movie production (Polyak et al., 2024; Zhao et al., 2025), virtual reality (Wu et al., 2025a;b), and digital human creation (Hu, 2024; Xiang et al., 2025; Zhang et al., 2025; Zhu et al., 2024). Despite significant advances in video synthesis, creating extended sequences with both temporal coherence and computational efficiency remains challenging (Ning et al., 2024).

Conventional diffusion-based methods (Peebles & Xie, 2023; Wang et al., 2025a; Chen et al., 2024b; 2023; Gupta et al., 2024; Ma et al., 2025) have achieved remarkable quality by jointly optimizing all frames via bidirectional attention. However, this global optimization necessitates the simultaneous generation of the entire sequence, introducing significant latency and rendering these methods impractical for real-time or interactive scenarios.

Autoregressive (AR) models (Wang et al., 2025b; Pang et al., 2025; He et al., 2024) offer an effective alternative by sequentially generating images or frames. This incremental strategy enables users to start viewing immediately after the initial frames are available, greatly reducing latency. Furthermore, AR models impose fewer constraints on video duration and facilitate interactive user control. Representative AR methods such as VideoGPT (Yan et al., 2021), LBD (Yu et al., 2024), and CogVideo (Hong et al., 2023) adopt a next-frame prediction paradigm based on discrete tokenizers, substantially lowering latency compared to diffusion-based approaches. However, their reliance on discrete tokenization inherently leads to quantization artifacts, reducing visual fidelity. Hybrid AR-diffusion methods (Sun et al., 2025; Chen et al., 2024a; Song et al., 2025) merge autoregressive generation with continuous diffusion processes to overcome these limitations. By integrating diffusion into the autoregressive framework, these methods avoid discrete codebooks, effectively addressing quantization-induced degradation and significantly improving output quality.

Nevertheless, both AR and AR-diffusion methods suffer from error accumulation. Since each frame depends explicitly on previously generated frames, errors from early frames compound and magnify over subsequent predictions, causing long-term degradation and temporary drift. Moreover, existing autoregressive approaches remain strictly sequential, inherently preventing parallel generation and thus limiting computational efficiency and scalability. These fundamental challenges motivate the question: *How can AR models move beyond naive autoregressive modeling to enable high-quality and parallelized long-video synthesis?*

Analogous to the workflow of professional filmmakers, long video creation naturally benefits from a hierarchical *plan-then-populate* paradigm. In a typical movie production, the process does not proceed by shooting every frame in chronological order. Instead, the production team first develops a Macro Plan, a rough storyboard that captures the overall structure and key moments of the film. This Macro Plan consists of multiple Micro Plans, each representing an individual scene or shot. With this setup, different scenes can be filmed in parallel according to their Micro Plans, much like multiple crews shooting on separate sets at the same time. The Macro Plan then coordinates and assembles all these pieces into a coherent long movie. Such hierarchical planning improves the efficiency of film production while ensuring that the final movie remains seamless and coherent. Building on this insight, we first perform a systematic analysis of error accumulation in AR and Non-AR video generation, revealing the fundamental mechanisms that drive long-term drift. Guided by these findings, we propose a novel plan-then-populate framework centered on Macro-from-Micro Planning (MMPL) for scalable, high-quality long video generation. MMPL operates via two complementary planning levels: *Micro Planning* efficiently predicts multiple keyframes of each segment simultaneously from its initial frame, capturing detailed local trajectories; *Macro Planning* autoregressively chains these segments by initializing each segment $S$ from the last keyframe of segment $S - 1$, thus ensuring global narrative coherence across the entire video. Once all keyframes are established, MMPL-based Content Populating concurrently synthesizes intermediate frames between keyframes within each segment, adhering to boundary constraints and eliminating sequential frame dependencies. To further optimize pipeline efficiency, we introduce an adaptive workload scheduling strategy that dynamically allocates GPU resources. This approach significantly reduces the overall generation time to approximately one-third of the original, without relying on distillation-based acceleration, while preserving high visual fidelity.

Overall, our work delivers the following contributions:

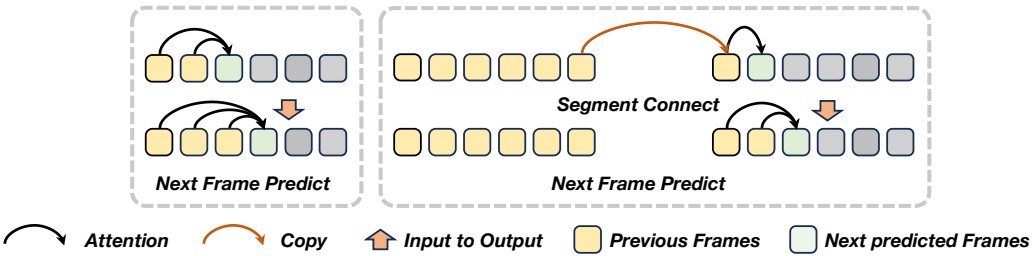

Figure 2: Existing AR methods generate frames sequentially in a step-by-step manner, inevitably causing error accumulation (as shown in Figure 1) and prohibiting parallel generation.

- We propose *Macro-from-Micro*, a hierarchical autoregressive planning method that forms coherent global storylines across segments of the entire video, while drastically reducing temporal error accumulation in long-video generation.
- We propose MMPL-based Content Populating, which synthesizes frames for multiple segments in parallel under the guidance of pre-planned keyframes, breaking the intrinsic sequential bottleneck of conventional autoregressive pipelines.
- We further design an adaptive multi-GPU workload scheduling strategy that balances segment generation across devices, substantially reducing wall-clock time for long-video synthesis.

## 2 RELATED WORK

**Bidirectional Diffusion Models for Video Generation.** Diffusion models have emerged as a dominant approach for high-quality visual synthesis, benefiting from their scalability and superior generative capabilities (Rombach et al., 2022; Dhariwal & Nichol, 2021). In video generation, existing diffusion architectures primarily rely on bidirectional attention mechanisms to jointly denoise all frames within a sequence (Guo et al., 2024; Ho et al., 2022; Blattmann et al., 2023; Huang et al., 2025b; Zhang et al., 2024). While this enables high-fidelity outputs, the requirement to concurrently generate entire sequences prohibits streaming or incremental video generation, resulting in significant inference latency and hindering applications involving long video generation.

**Causal Autoregressive Models for Video Generation.** Autoregressive (AR) models (Sun et al., 2025; Chen et al., 2024a; Song et al., 2025; Deng et al., 2025; Gao et al., 2024; Li et al., 2025) provide an alternative by sequentially generating video frames or spatiotemporal tokens, conditioning each new frame on previously generated content as shown in Figure 2. This causal generation paradigm naturally supports streaming outputs and substantially reduces initial latency. However, the sequential dependency between frames inherently introduces error accumulation. As prediction chains grow longer, these errors compound, resulting in temporal drift and degraded visual coherence, especially noticeable in extended video sequences.

**Methods for Long Video Generation.** Long video synthesis poses unique challenges due to cumulative errors and computational bottlenecks inherent in autoregressive inference. Recent efforts, such as CausVid (Yin et al., 2025) and Self Forcing (SF) (Huang et al., 2025a), address these challenges by introducing methods like *Diffusion Forcing* and *Self Forcing*, aimed at reducing the mismatch between training and inference dynamics. Although these techniques partially alleviate drift through recursive conditioning and short-step diffusion, they remain susceptible to significant error propagation when generating videos exceeding approximately 30 seconds.

**Planning Prediction.** A closely related work, FramePack-Plan (Zhang & Agrawala, 2025), mitigates error accumulation via step-wise frame jumping, and compresses context to extend video length. In contrast, our Macro-from-Micro framework introduces three key innovations. First, we adopt a two-level hierarchical planning scheme: a Micro Plan predicts segment-level keyframes, and a Macro Plan, composed of overlapping Micro Plans, forms a coherent global storyline through autoregressive scheduling. Second, each Micro Plan produces all pre-planned keyframes for its segment in a single forward pass conditioned only on the initial frame, drastically compressing the autoregressive chain. Finally, once the Macro Plan is obtained, the remaining content within all segments is synthesized in parallel, achieving high throughput while preserving temporal coherence.

Figure 3: Overall framework of Macro-from-Micro Planning. Our method operates on two planning levels: (1) Micro Planning, which predicts a sequence of future frames inside its own segment to mitigate local error accumulation, and (2) Macro Planning, formed as an Autoregressive Chain of Micro Plans, where the planning frames of the first segment autoregressively generate the planning frames of subsequent segments, ensuring long-horizon temporal consistency.

## 3 METHOD

### 3.1 MACRO-FROM-MICRO PLANNING

Motivated by the analysis in the supplementary material, we observe that autoregressive models accumulate errors proportionally to the number of propagation steps, whereas non-autoregressive models decouple errors from the step count through joint optimization. To exploit the complementary strengths of both paradigms, we introduce *Macro-from-Micro Planning (MMPL)*, a unified planning method comprising two key components: *Micro-Planning* and *Macro-Planning*.

**Micro Planning.** Micro Planning $\mathcal{M}_s$ constructs a short temporal storyline for the $s$-th segment with $N$ frames by predicting a small set of key frames, denoted as $\mathcal{P}_{\mathcal{M}_s}$, that act as stable anchors for subsequent content synthesis. This sparse set of *pre-planning frames*, $\{x_s^{t_a}, x_s^{t_b}, x_s^{t_c}\}$, is jointly predicted from the initial frame $x_s^1$. This process can be expressed as:

$$p(\mathcal{P}_{\mathcal{M}_s} \mid x_s^1) = p(x_s^{t_a}, x_s^{t_b}, x_s^{t_c} \mid x_s^1). \tag{1}$$

Where $t_a = 2$ denotes the early neighbor of the initial frame, $t_b = N/2$ serves as the global midpoint, and $t_c = N$ marks the terminal frame of the sequence. These *pre-planning frames* are jointly optimized while conditioned solely on the initial frame $x_s^1$, rendering their mutual drift with $x_s^1$ negligible. Moreover, since all frames are jointly optimized from the initial frame $x_s^1$, their residual errors are mutually constrained and remain negligible, preventing the cumulative drift characteristic of sequential autoregressive generation. This design ensures temporal coherence within each segment and establishes a stable, drift-resistant foundation for the subsequent populating process.

**Macro Planning.** While Micro Planning provides a segment-level temporal storyline, it remains limited in capturing global dependencies across the entire video. To achieve long-range coherence, we extend Micro Planning into *Macro Planning*, denoted as $\mathcal{M}^+$. Macro Planning constructs a global storyline for the entire long video by sequentially chaining overlapping Micro Plannings across video segments. Concretely, the terminal pre-planning frames of one segment serve as the initial conditions for the next, thereby linking local plans into a coherent long-horizon structure, which can be regarded as a segment-level autoregressive process over the video timeline. Let the full video of frame length $T$ be partitioned into $S$ short segments, with the initial frame of the $s$-th segment denoted as $x_s^1$. Let the set of predicted planning frames produced by Macro Planning be denoted as $\mathcal{P}_{\mathcal{M}^+}$. This process can be expressed as:

$$p(\mathcal{P}_{\mathcal{M}^+} \mid x_1^1) = \prod_{s=1}^{S} p(\mathcal{P}_{\mathcal{M}_s} \mid x_s^1), \quad x_{s+1}^1 := x_s^{t_c}, \quad \mathcal{P}_{\mathcal{M}^+} := \bigcup_{s=1}^{S} \mathcal{P}_{\mathcal{M}_s}. \tag{2}$$

where $\mathcal{M}_s$ represents the Micro Planning for the $s$-th segment. By hierarchically chaining these segment-level plans, Macro Planning transforms the original frame-by-frame long-range autoregressive dependency into a segment-wise sequence of sparse planning dependencies. This restructuring

preserves global temporal coherence by ensuring a consistent storyline across segments and suppresses temporary drift, effectively reducing the error accumulation scale from the $T$-frame level of conventional autoregressive generation to the $S$-segment level under our framework, where $S \ll T$.

However, when linking Micro Plannings through an autoregressive chain, directly reusing the tail latent tokens of the preceding segment as the prefix for the next often leads to boundary flickering and color shifts across segment transitions. This issue stems from a distribution mismatch. The first latent frame fundamentally differs from the others: it represents only the initial image, while subsequent frames incorporate temporally compressed information, resulting in inconsistent statistics across frames. Therefore, inspired by CausVid (Yin et al., 2025), we introduce a drift-resilient re-encoding and decoding strategy to stabilize inter-segment transitions. Specifically, as shown in Figure 4, we first concatenate the initial latent token of the preceding segment with its terminal planning tokens and feed the sequence into the VAE decoder for video reconstruction. However, since VAE decoding requires each token to condition on strictly contiguous temporal prefixes, any temporal discontinuity in the input sequence leads to pronounced color shifts and boundary artifacts. To mitigate this issue, we duplicate the terminal planning tokens once and insert the copy between the initial latent token and the original terminal planning tokens, forming a temporally contiguous latent sequence for decoding. After reconstruction, we re-encode the second copy

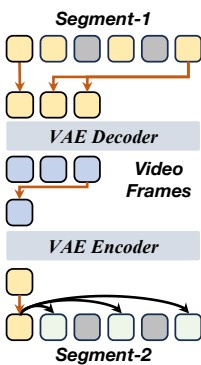

Figure 4: Our Re-Encoding and Decoding Strategy.

of the terminal planning tokens and use the resulting latents as the initial tokens for the next segment's Micro Planning. This design enforces both statistical and temporal consistency in the latent space, effectively suppressing color shifts and boundary flickering, and achieving smooth, stable inter-segment transitions.

### 3.2 MMPL-BASED CONTENT POPULATING

Following Sec. 3.1, the Micro Plan $\mathcal{M}_s$ naturally partitions each video segment into two *sub-segments*, bounded by consecutive planning frames, e.g., $\left[x_s^{t_a}, x_s^{t_b}\right]$ and $\left[x_s^{t_b}, x_s^{t_c}\right]$. To synthesize the complete segment by populating the remaining frames under the constraints of these planning frames, we introduce MMPL-based Content Populating. Specifically, Micro Planning generates three types of planning frames: *early*, *midpoint*, and *terminal*. Inspired by early methods that generate videos conditioned on the first and last frames, we divide the Content Populating process into two stages, as shown in Figure 5. In the first stage, we populate the interval by using the initial and early planning frames as the head and the midpoint planning frames as the tail, synthesizing the intermediate content. In the second stage, we extend the populated sequence by taking all frames between the initial frame and the midpoint planning frames as the new head and the terminal frames as the tail, thereby generating the remaining content. This process can be expressed as:

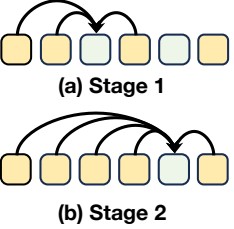

Figure 5: Two Stages of our MMPL-based Content Populating.

$$p(\mathcal{C}_s \mid \mathcal{P}_{\mathcal{M}_s}) = p\big(x_s^{t_a+1:t_b-1} \mid x_s^{1:t_a}, x_s^{t_b}\big) \cdot p\big(x_s^{t_b+1:t_c-1} \mid x_s^{1:t_b}, x_s^{t_c}\big), \tag{3}$$

where $\mathcal{C}_s$ corresponds to the content frames to be synthesized in the $s$-th segment. The variables $x_s^{t_a}$, $x_s^{t_b}$, and $x_s^{t_c}$ denote the early, midpoint, and terminal planning frames of segment $s$, respectively. The notation $x_s^{1:t_a}$ and $x_s^{1:t_b}$ indicates that the generation of each sub-segment is conditioned not only on its boundary planning frames but also on all preceding frames in the same segment. Accordingly, the intermediate frames within the two sub-segments, denoted as $x_s^{t_a+1:t_b-1}$ and $x_s^{t_b+1:t_c-1}$, represent the remaining content to be populated.

Specially, the factorization in Eq. 3 explicitly demonstrates that content population within each sub-segment depends exclusively on its corresponding planning frames. Consequently, multiple sub-segments can be independently optimized in parallel, provided their internal planning frames have been generated. Furthermore, leveraging multiple GPUs, the proposed MMPL-based Content Populating can distribute segment-wise optimization across different devices, enabling concurrent execution. This parallelization significantly enhances computational efficiency, facilitating highly efficient long-video synthesis.

Figure 6: Multi-GPU parallel inference via adaptive workload scheduling. Given the initial frame $f_1^0$, segment 0 first generates its planning frames $f_2^0$, $f_6^0$, and $f_{10}^0$. These planning frames then guide the content population of the intermediate frames $f_3^0$, $f_4^0$, and $f_5^0$. While segment 0 is still populating these frames, segment 1 can immediately start its Micro Planning by taking $f_{10}^0$ as the initial frame $f_1^1$ and generating its own planning frames $f_2^1$, $f_6^1$, and $f_{10}^1$. This staged execution enables overlapping planning and populating across segments, maximizing multi-GPU parallelism. Here, each $t_i$ denotes an inference step in the diffusion sampling process.

## 3.3 ADAPTIVE WORKLOAD SCHEDULING

As discussed in Sec. 3.2, the content populating of different segments can be executed in parallel across multiple GPUs. However, this approach suffers from a key limitation: parallelization cannot start until the planning frames of all segments have been fully generated, introducing an inevitable prefix delay that degrades the overall pipeline throughput. To further improve generation efficiency, we propose an *adaptive workload scheduling* strategy, which dynamically adjusts the execution order of Micro Planning, Macro Planning, and Content Populating to maximize parallelism. Specifically, Macro Planning is constructed as an autoregressive chain of segment-level Micro Plannings, which naturally enforces a strict generation order of planning frames across segments. This property allows us to initiate the Content Populating of an earlier segment as soon as its planning frames are available, without waiting for the planning frames of all subsequent segments to finish. To illustrate the workload scheduling, consider a case where we set $t_a = 2$, $t_b = 6$, and $t_c = 10$ to evenly cover the temporal span. As shown in Figure 6, the planning frames of the current segment, generated via *Micro Planning* ($x_s^{t_b}$ or $x_s^{t_c}$), immediately serve as the initial frame $x_{s+1}^1$ for the subsequent segment. This allows the next segment to start its own *Micro Planning* while the current segment is still performing *Content Populating* to generate $x_s^{t_a+1:t_b-1}$. This staged independence naturally enables segment-parallel generation, as formally expressed in Eq. (4):

$$\begin{aligned}
\text{Segment s:} \quad & x_s^{t_a+1:t_b-1} \sim p_\theta(x \mid x_s^1, x_s^{t_a}, x_s^{t_b}), \\
\text{Segment s+1:} \quad & \{x_{s+1}^{t_a}, x_{s+1}^{t_b}, x_{s+1}^{t_c}\} \sim p_\theta(x \mid x_{s+1}^1), \quad x_{s+1}^1 \in \{x_s^{t_b}, x_s^{t_c}\}.
\end{aligned} \tag{4}$$

Here, the initial frame $x_{s+1}^1$ of the next segment can be selected either as $x_s^{t_b}$ or $x_s^{t_c}$. This selection directly determines the parallel execution strategy and leads to two distinct modes:

**(1) Minimum Memory Peak Prediction.** When $x_s^{t_b}$ is used as $x_{s+1}^1$, intermediate frames $x^{t_b+1}$ : $x^{t_c-1}$ are skipped, bypassing the region with the deepest temporal context and highest generation latency. This mode minimizes peak memory usage and reduces per-segment latency but introduces frame reuse between segments, slightly reducing overall throughput.

**(2) Maximum Throughput Prediction.** When $x_s^{t_c}$ is used as $x_{s+1}^1$, all intermediate frames are generated sequentially within the segment, eliminating inter-segment redundancy and achieving maximal pipeline efficiency, at the cost of higher per-segment computation.

These two execution strategies offer a trade-off between local memory/latency and global throughput, allowing flexible deployment choices.

Table 1: Evaluation metrics for the other baselines and MMPL. The first five metrics are automatically computed by VBench, while the last three are obtained through human evaluation.

| Model | VBench-long Evaluation | | | | | Human Evaluation | | |
|---|---|---|---|---|---|---|---|---|
| | Subject Consistency | Background Consistency | Motion Smoothness | Aesthetic Quality | Imaging Quality | Text-Visual Alignment | Content Consistency | Color Shift |
| *Causal* | | | | | | | | |
| FIFO (Kim et al., 2024) | 0.956 | 0.960 | 0.949 | 0.588 | 0.603 | - | - | - |
| *Distilled Causal* | | | | | | | | |
| CausVid((Yin et al., 2025)) | 0.969 | **0.980** | 0.981 | 0.606 | 0.661 | 34.7 | 33.0 | 25.0 |
| SF (Huang et al., 2025a) | 0.967 | 0.958 | 0.980 | 0.593 | **0.689** | 52.0 | 46.1 | 50.5 |
| *DF Causal* | | | | | | | | |
| SkyReels (Chen et al., 2025) | 0.956 | 0.966 | 0.991 | 0.600 | 0.581 | 47.9 | 51.4 | 51.3 |
| MAGI-1 (Teng et al., 2025) | 0.979 | 0.970 | 0.991 | 0.604 | 0.612 | 34.7 | 40.4 | 39.5 |
| *Planning* | | | | | | | | |
| MMPL-1.3B | **0.980** | 0.970 | 0.987 | 0.600 | 0.665 | - | - | - |
| MMPL-14B | **0.980** | 0.968 | **0.992** | **0.628** | 0.661 | **80.0** | **79.2** | **83.1** |

# 4 EXPERIMENTS

**Baselines.** We compare our model against representative open-source video generation systems of comparable scale, including FIFO (Kim et al., 2024), SkyReelsV2 (Chen et al., 2025), MAGI (Teng et al., 2025), CausVid (Yin et al., 2025), and Self Forcing (Huang et al., 2025a). All methods are evaluated under a unified sliding-window protocol, where each fixed-length segment (e.g., 5 s) is causally conditioned on the final frames of the preceding segment. We adopt SkyReels-V2-14B and MAGI-4.5B as our primary baselines, while CausVid and Self Forcing (1.3B, distilled from 14B teachers) serve as high-fidelity autoregressive representatives.

**Training Details.** We implement *MMPL* on both the 1.3B and 14B variants of Wan2.1-T2V (Wang et al., 2025a), a DiT-based (Peebles & Xie, 2023) Flow Matching model originally built for 5-second videos. To support long-horizon modeling, we adopt FlexAttention (Dong et al., 2024) for scalable training and FlashAttention-v3 (Dao et al., 2022) for fast inference. The model is fine-tuned on 50k curated high-quality videos at $832 \times 480$ resolution, ensuring diverse and clean data for stable optimization. Training runs for 8,000 iterations on 32 H100 GPUs with AdamW at a $1 \times 10^{-5}$ learning rate. For hierarchical planning, we set $t_a = 2, 3$, $t_b = 10, 11, 12$, and $t_c = 19, 20$, corresponding to early, midpoint, and late planning frames guiding segment-wise generation. Additional hyperparameters and ablation settings are provided in the supplementary material.

**Evaluation.** We evaluate on the VBench-long benchmark (Zheng et al., 2025), which assesses subject and background consistency, motion smoothness, aesthetic quality, and imaging quality, jointly reflecting temporal stability and perceptual fidelity. For the main study, we generate 30s videos from 120 randomly sampled MovieGen (Polyak et al., 2024) prompts on a single H100 GPU. To complement these quantitative metrics, we also conduct a user study: for each baseline, 19 videos of about 30 s are generated using the first 19 MovieGen prompts, and 29 participants perform pairwise comparisons, selecting the video that better matches the prompt in terms of visual quality and semantic fidelity. This combination of objective metrics and human judgments provides a rigorous evaluation of both numerical performance and perceptual quality. Details of the user study are provided in the supplementary material.

**Quantitative Results.** As shown in Table 1, both our 1.3B and 14B models achieve strong overall performance on VBench-long, with the 14B model excelling in subject consistency 0.980, motion smoothness 0.992, and aesthetic quality 0.628, while maintaining competitive imaging quality 0.661 and only slightly lower background consistency 0.968 than CausVid and MAGI-1. However, VBench metrics, particularly subject and background consistency, tend to favor less dynamic scenes and cannot fully capture the perceptual complexity of long video generation. To address this limitation, we conducted a human study by generating 19 diverse 30-second videos per method, spanning humans, vehicles, and natural landscapes. Thirty participants rated each video on text-visual alignment, content consistency, and long-sequence color stability. Our method achieved the highest scores in all three dimensions: 80.0 for text-visual alignment, 79.2 for content consistency, and 83.1 for color stability, substantially outperforming other baselines. Besides, as illustrated in Figure 1, our method is consistently preferred in human evaluations, confirming its perceptual advantage.

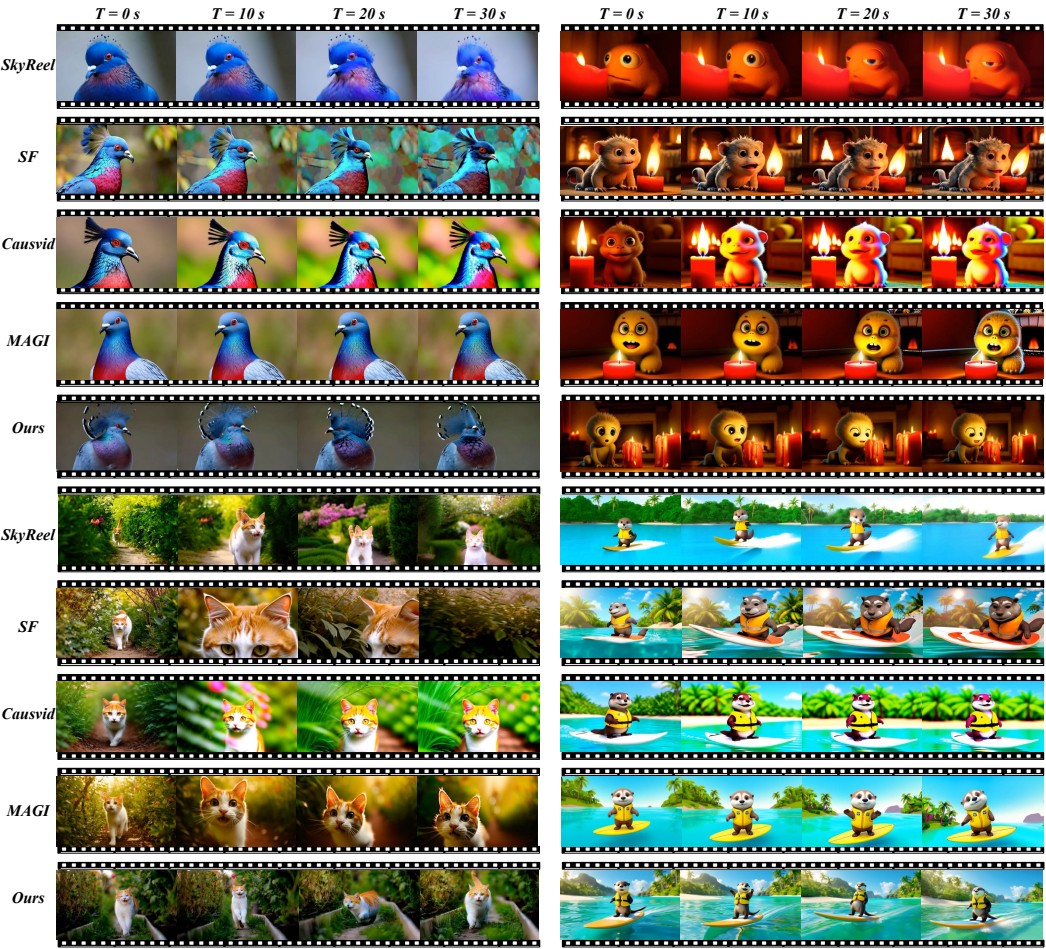

Figure 7: Qualitative comparisons. We visualize videos generated by Macro-from-Micro against those by MAGI, SkyReels-V2, Self Forcing, and CausVid.

**Qualitative Results.** As illustrated in Figure 7, AR base-lines exhibit severe temporal drift, caused by error accumulation during long-video generation. Over the course of 30-second sequences, these models progressively lose visual fidelity, with artifacts such as blurring, fading, and noticeable color drift becoming increasingly pronounced. The degradation often compounds in dynamic scenes, where motion discontinuities and geometric distortions further undermine temporal coherence. In contrast, our approach sustains high visual quality across the entire sequence, demonstrating strong robustness to motion drift and color distortion. It consistently surpasses CausVid and Self Forcing, and further achieves superior performance to SkyReels-V2 and MAGI-1 under challenging long-horizon conditions, highlighting its effectiveness for stable and high-fidelity long video synthesis.

**Parallel Inference Efficiency.** To highlight the practical advantages of Macro-from-Micro Planning, we compare its standard inference with the parallelized variant. The

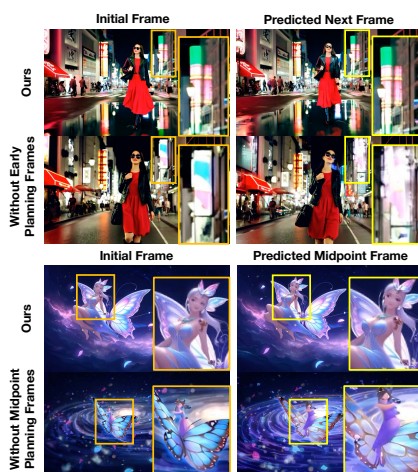

Figure 8: Qualitative comparisons of different MMPL variants.

parallel strategy achieves substantial speedups without compromising generation quality. As illustrated in Figure 1, our method significantly reduces generation time for 60-second videos, demon-

Table 2: Ablation studies on planning setups.

| Variant | VBench | | | | |
|---|---|---|---|---|---|
| | Subj. | Back. | Mot. | Aes. | Img. |
| *Planning Setup* | | | | | |
| w/o early planning | 0.972 | 0.964 | 0.991 | 0.610 | 0.640 |
| w/o midpoint planning | 0.977 | 0.968 | 0.992 | 0.618 | 0.637 |
| Full | 0.980 | 0.968 | 0.992 | 0.628 | 0.661 |

Table 3: Ablation studies on training strategies.

| Variant | VBench | | | | |
|---|---|---|---|---|---|
| | Subj. | Back. | Mot. | Aes. | Img. |
| *Training Strategy* | | | | | |
| Freeze | 0.838 | 0.923 | 0.973 | 0.484 | 0.503 |
| Only Q,K | 0.970 | 0.962 | 0.987 | 0.612 | 0.647 |
| Only Self-Attention | 0.980 | 0.968 | 0.992 | 0.628 | 0.661 |

strating strong scalability and suitability for real-time deployment. Notably, using only two GPUs halves the inference time, and thanks to the pipeline design, four GPUs further reduce the generation time to roughly one-third of the original. These results confirm that our approach effectively balances throughput and quality, and its hardware efficiency makes it highly amenable to large-scale video synthesis applications.

**Ablations on Micro-Planning Frame Placement.** The placement of Planning frames within each segment during *Micro Planning* is pivotal for MMPL, shaping temporal and structural consistency. We validate this via an ablation with three variants: (i) without early frames (omit frames near the start); (ii) without the midpoint frame (remove the central anchor); and (iii) the full MMPL that retains all Planning frames. As shown in Table 2, the full configuration leads across all metrics. Qualitatively (Figure 8), it yields smoother transitions and more stable long-horizon content, whereas the ablated variants exhibit discontinuities or noticeable jumps around the missing frames.

**Ablations on Model Training Strategy.** We compare three update policies for the video generation model: Freeze freezes all parameters; Only Q,K updates only the self-attention query and key projections; Only Self-Attention updates Q, K, V and the attention output, while feed-forward layers and embeddings remain frozen. As shown in Table 3, updating the whole self-attention yields the best scores across all metrics. Training only Q,K is lighter but slightly weaker. Freezing performs worst and shows larger temporal drift and inconsistency.

**Performance on Short-Horizon Generation.**
Although MMPL is primarily designed for long-horizon video synthesis, it also improves short-horizon generation quality. We compare MMPL with the strong baseline Wan-2.1-14B on 5-second clips using VBench metrics. As shown in Table 4, MMPL achieves slightly higher perceptual scores across subject consis-

Table 4: Comparison on 5-second videos.

| Model | Subj. | Back. | Mot. | Aes. | Img. |
|---|---|---|---|---|---|
| Wan-2.1 | 0.980 | 0.970 | 0.988 | 0.600 | 0.671 |
| **MMPL** | **0.984** | **0.971** | **0.993** | **0.629** | 0.663 |

tency, background consistency, motion smoothness, and aesthetics, demonstrating that our planning mechanism benefits both short- and long-range video generation.

## 5 EXTENDED EXPERIMENTAL ANALYSIS

Although VBench provides a widely adopted protocol for benchmarking video generation models, its global and frame-wise metrics are not always sensitive to localized or long-horizon degradations. To provide a more complete understanding of MMPL's behavior on long sequences, we introduce complementary evaluations that capture both color drift and temporal distribution consistency.

**Color-Shift Metrics** We additionally measure long-range color stability using frame-to-frame hue statistics. Specifically, we compute the H-channel L1 distance and H-channel correlation between the first and last frame of each 30-second video, providing a direct estimate of global color drift. As shown in Table 5, MMPL achieves the smallest drift and highest hue consistency among all methods.

**Long-Video Consistency via JEPA Metrics** To evaluate temporal coherence over extended horizons, we adopt the JEPA-based metrics (Balestriero et al., 2025). Although JEPA was originally designed to estimate dataset-level distributions, it can be naturally repurposed for long video analysis by treating a single video as a dataset and each frame as a data sample. Under this reinterpretation, taking the first frame as the reference distributional center allows the JEPA score standard deviation to quantify temporal drift, while the first–last frame score difference measures the accumulated bias over time. Table 6 shows that MMPL achieves the most stable temporal distribution, exhibiting both the lowest JEPA variance and the smallest long-range frame discrepancy.

Table 5: Color-shift metrics computed between the first and last frames.

| Variant | H-channel L1 Distance ↓ | H-channel Correlation ↑ |
|---|---|---|
| CausVid | 0.711 | 0.598 |
| Self-Forcing | 1.152 | 0.162 |
| FramePack | 0.445 | 0.881 |
| **MMPL** | **0.306** | **0.927** |

Table 6: JEPA-based long-video consistency metrics.

| Model | JEPA-Score Std ↓ | First–Last Score Diff ↓ |
|---|---|---|
| CausVid | 0.1107 | 0.4093 |
| Self-Forcing | 0.2293 | 1.0695 |
| FramePack | 0.0853 | 0.1364 |
| **MMPL** | **0.0705** | **0.0281** |

Table 7: Comparison with planning-based baselines under matched model scales. Lower JEPA-std and JEPA-Diff indicates better long-horizon stability.

| Model | Params | Subj. | Back. | Mot. | Aes. | Img. | JEPA-Std | JEPA-Diff |
|---|---|---|---|---|---|---|---|---|
| FramePack | 13B | 0.987 | 0.971 | 0.996 | 0.607 | 0.638 | 0.0853 | 0.1364 |
| MMPL-14B | 14B | 0.980 | 0.968 | 0.992 | **0.628** | **0.661** | **0.0705** | **0.0281** |

**Comparison with Other Planning Methods** We compare MMPL with planning-based baselines under matched model scales. Since FramePack-Plan is not publicly available and the released FramePack code supports only image-to-video inference, we adopt a practical setting by feeding FramePack with the first frame generated by our method to ensure comparable initialization. As shown in Table 7, MMPL-14B achieves the strongest overall performance, with significantly better aesthetic quality, imaging quality, and much lower temporal drift, while the remaining consistency metrics are at a similar level to FramePack. These results highlight the clear advantage of our planning mechanism in maintaining stable and coherent long-horizon video generation.

## 6 DISCUSSION

**Compatibility with Acceleration and Distillation Methods.** Our paradigm is naturally compatible with acceleration techniques such as DMD and other distillation approaches, requiring no architectural changes. During training, we only adjust the attention mask to control the visible frame range, while at inference efficiency is improved by reorganizing the decoding order of video segments. This flexibility allows Macro-from-Micro to plug into existing acceleration pipelines. The results of adapting model distillation and related strategies are provided in the supplementary material.

**Compatibility with Self-Forcing Approaches.** MMPL also complements self-correcting strategies that mitigate step-wise autoregressive errors, such as Self Forcing. In standard training, the model predicts the next frame by denoising conditioned on ground-truth video frames; replacing these with previously generated predictions naturally yields a Self-MMPL regime. This hybrid setup extends the duration of generable videos and improves temporal consistency over long sequences.

**Limitations.** Although Macro-from-Micro Planning substantially mitigates the accumulation of prediction errors, generating hour-level videos remains challenging due to the reliance on a single static text prompt. The substantial temporal span of long videos means that a single text prompt often aligns only with the early content and fails to capture the full video semantics. Since our current framework does not dynamically update prompts during generation, this static conditioning limits the ability to produce coherent and continuous content across longer segments.

## 7 CONCLUSION

In this work, we propose a novel planning-then-populating framework centered on Macro-from-Micro Planning (MMPL) for long video generation, which operates without modifying the underlying model architecture. By decomposing video synthesis into Micro Planning, Macro Planning, and Content Populating, MMPL significantly mitigates temporal drift, ensures long-term consistency, and unlocks substantial parallelism in frame generation. Combined with Adaptive Workload Scheduling, it further accelerates long-video generation, reducing inference time to nearly one-third of the original without distillation techniques. Extensive quantitative and qualitative experiments validate the superior performance of our approach. In the future, we plan to integrate MMPL with model distillation techniques to enable real-time, fully parallelized long video generation, making the framework highly practical for interactive and streaming applications.

ETHICS STATEMENT

All authors have read and agree to abide by the ICLR Code of Ethics. This work does not involve interventions with human participants or personally identifiable information. We use only publicly available datasets under their original licenses and follow the terms of use. Potential risks and our mitigations are summarized below:

- **Privacy & Security.** We do not collect or release any personal data. When showing qualitative examples, all images/videos are from public datasets; any sensitive content is filtered.
- **Bias & Fairness.** We report results on multiple benchmarks and provide detailed settings to facilitate external auditing. We acknowledge possible dataset biases and encourage follow-up evaluation on broader demographics and domains.
- **Dual Use / Misuse.** The method could be misused to enable undesired large-scale labeling or surveillance. To reduce misuse, we release only research artifacts (code/configs) and exclude any tools for scraping or re-identifying individuals.
- **Legal Compliance.** We comply with licenses of all third-party assets (code, models, and datasets) and cite their sources. Any additional third-party terms are respected.
- **Research Integrity.** We document preprocessing, training recipes, and evaluation protocols; random seeds and hyperparameters are provided to enable reproducibility.

Where applicable, institutional review information is withheld for double-blind review and can be provided after acceptance.

REPRODUCIBILITY STATEMENT

We include training and evaluation details in the main paper and Appendix. Concretely: (i) all hyperparameters, optimization settings, and compute budgets; (ii) full data preprocessing and splits. Complete code and training logs will be open-sourced upon paper acceptance.

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

## A  DATA PREPARATION

**Data Sources and Filtering.** Our dataset comprises two components: (i) licensed commercial videos purchased from authorized providers; and (ii) web videos manually collected from open platforms, primarily Mixkit, Pexels, and Pixabay. For all candidate videos, we first generate textual captions with Qwen-72B and compute an aesthetic score using the LAION aesthetic predictor. For licensed data, we rank by the aesthetic score and retain only the top 1%. For web-collected data, we conduct human quality control to remove low-quality clips, content–caption mismatches, and potential copyright risks, and to verify caption consistency. We then merge the two subsets, yielding approximately ∼50k high-quality samples to train our long-video generation model.

**Structured Annotation Pipeline.** To obtain rich and structured annotations, we drive Qwen-72B with carefully designed instruction prompts to analyze video frames and output a JSON object with a fixed schema. The JSON includes: a short scene summary (`short_caption`); a dense contextual description (`dense_caption`, covering main subject, background, visual style, camera movement, shot type, lighting, and atmosphere); detailed subject descriptions (for persons: facial expressions, emotional state, and ethnicity); background information; standardized style/shot/movement labels; aesthetic tags; and role statistics (e.g., number of humanoid characters, coverage extent, depiction style, and motion dynamics). Concretely, `short_caption` is generated with the instruction *"Brief scene summary in 1 sentence"*, while `dense_caption` uses *"Detailed context including main subject, background, visual style, camera movement tech, shot type, lighting, and atmosphere"*. All outputs are in English, follow predefined field orders and constraints, and employ standardized vocabulary for key attributes.

**Quality Control and Training Setup.** After annotation, we re-evaluate aesthetic quality with the LAION predictor to ensure consistency. During training, for each of the ∼50k videos we sample the conditioning text with probability 0.8 from `dense_caption` and with probability 0.2 from `short_caption`. This strategy preserves the high information density of dense captions while maintaining robustness and diversity from concise summaries. As shown in Figure 9, we present several representative examples from our curated dataset.

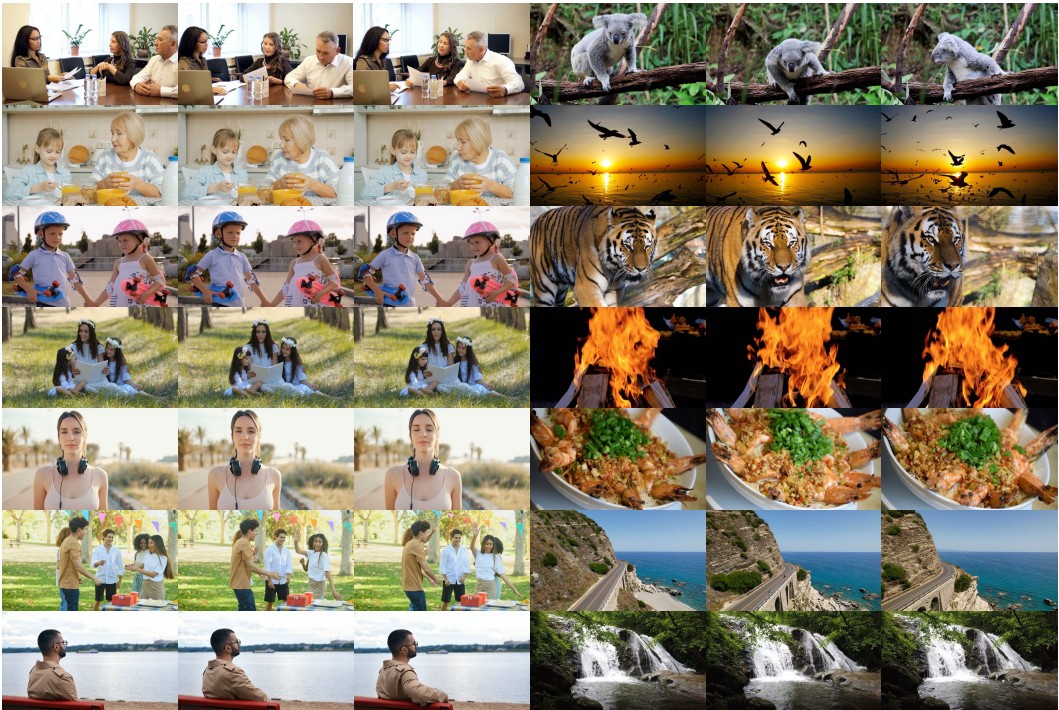

Figure 9: **Examples of training samples.** The dataset combines licensed and web-collected videos, curated via aesthetic scoring and manual screening.

# B    USER STUDY DETAILS

To complement the quantitative metrics, we conduct a user study on long-video generation. In each trial, participants evaluate five videos generated from the same prompt by ranking them (1 = best, 5 = worst) along three dimensions: *text–visual alignment*, *content consistency*, and *long-sequence color stability*. In addition, participants select a single overall favorite video.

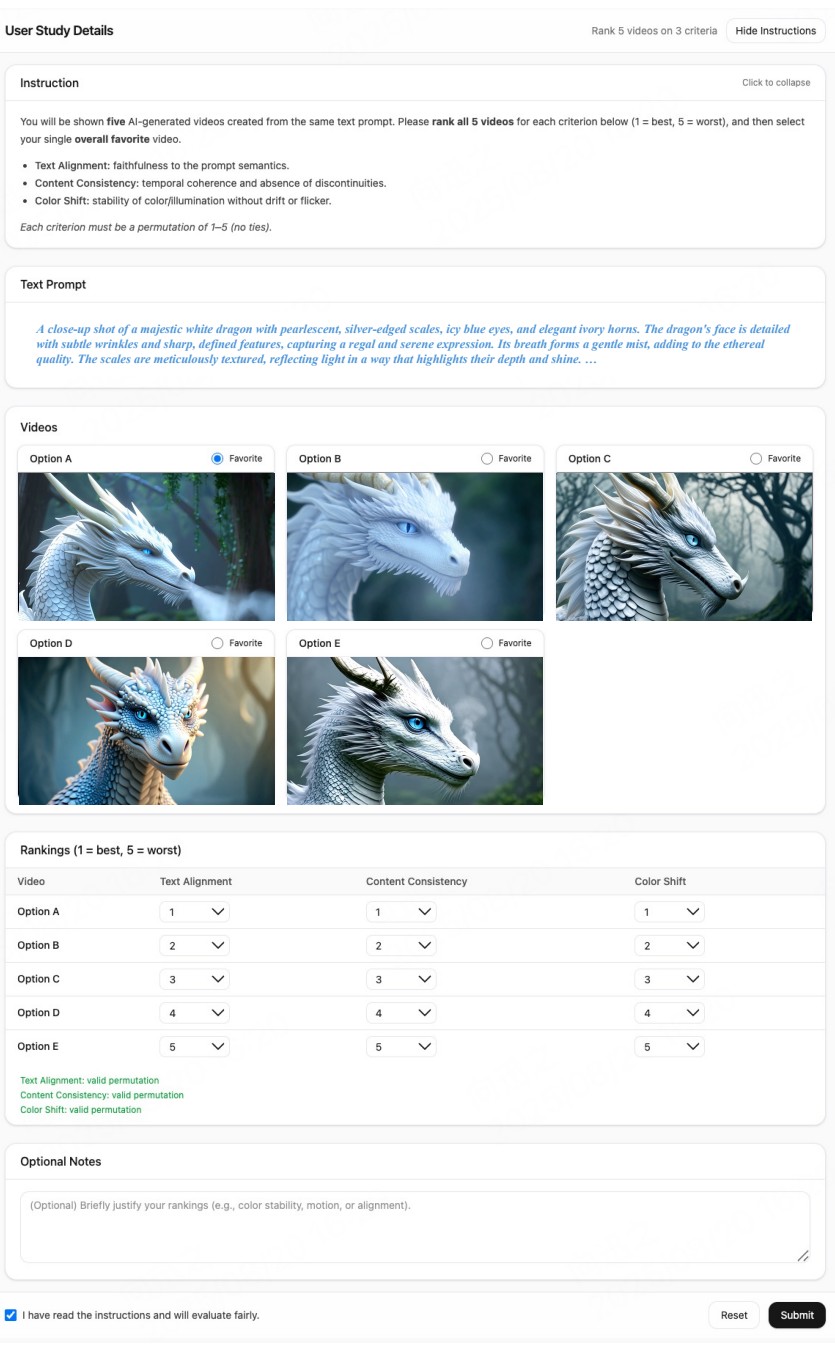

Figure 10: **User study instruction screenshots.**

This protocol provides fine-grained human judgments on both quality and temporal robustness that are not fully captured by automated metrics. Detailed instructions are shown in Figure 10.

## C TRAINING SETTINGS

### C.1 HYPERPARAMETER SETTINGS

Most experiments are conducted on 32 NVIDIA GPUs (80 GB each), using a per-GPU batch size of 1 without gradient accumulation. The detailed hyperparameters are summarized in Table 8. Training the Teacher Forcing 14B model for 8,000 steps required about three days, while the DMD 1.3B model reached 8,000 steps within roughly one day.

Table 8: Specification of training hyperparameters

| Hyperparameters | Teacher Forcing | Self Forcing |
|---|---|---|
| Generate network | Wan2.1-T2V-14B | Wan2.1-T2V-1.3B |
| Real score network | N/A | Wan2.1-T2V-14B |
| Fake score network | N/A | Wan2.1-T2V-14B |
| Batch size | 32 | 32 |
| Optimizer ($G_\theta$) | AdamW, $\beta_1 = 0$, $\beta_2 = 0.999$, $\epsilon = 1 \times 10^{-8}$, weight_decay $= 0.01$ | Adam, $\beta_1 = 0$, $\beta_2 = 0.999$, $\epsilon = 1 \times 10^{-8}$, weight_decay $= 0.01$ |
| Optimizer ($f_\psi$) | N/A | Adam, $\beta_1 = 0$, $\beta_2 = 0.999$, $\epsilon = 1 \times 10^{-8}$, weight_decay $= 0.01$ |
| Learning rate ($G_\theta$) | $1 \times 10^{-5}$ | $2 \times 10^{-6}$ |
| Learning rate ($f_\psi$) | N/A | $4 \times 10^{-7}$ |
| Gen./Cri. update ratio | N/A | 5 |
| EMA decay | N/A | 0.99 |

### C.2 PLANNING SETTINGS

**Settings.** To clarify the generation process, we detail the model's computation using the 21 latent tokens of the full Wan model as shown in Figure 11. Tokens are indexed from 0: indices 0–1 correspond to the initial frame, indices 2–3 to early planning frames, indices 10–12 to midpoint planning frames, and indices 19–20 to terminal planning frames.

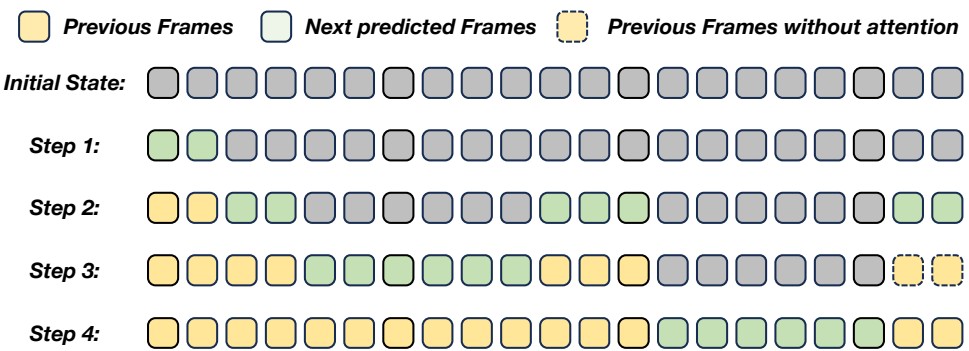

Figure 11: **Overview of our planning-based inference on an 81-frame sequence with 21 latent tokens (full Wan model).** Pre-Planning latent tokens at the beginning, midpoint, and terminal positions serve as stable anchors in the denoising schedule, guiding the synthesis of all intermediate frames and ensuring long-range temporal coherence.

**Analysis.** While the proposed planning setup places anchors over long horizons, a central challenge remains: enabling the autoregressive (AR) decoder to effectively exploit these anchors when synthesizing intermediate frames. Relying on a single planning token at the *early*, *midpoint*, and *terminal* boundaries is intrinsically fragile in the presence of the AR decoder's pronounced *recency bias*—the tendency to overweight the most recent observations while underutilizing distant context. This bias causes the model, at each sub-segment junction, to condition predominantly on the tail of the preceding sub-segment, thereby inheriting and amplifying residual errors and inducing cross-boundary propagation. Consequently, a single planning token per boundary is insufficient to arrest drift arising from accumulated errors. Formally, this bias in MMPL is expressed in Eq. 5:

$$p\big(x^{s_k+1:e_k-1} \mid x^{1:s_k}, x^{e_k}\big) \approx p\big(x^{s_k+1:e_k-1} \mid x^{s_k-K:s_k}, x^{e_k}\big). \tag{5}$$

Here, $s_k$ and $e_k$ denote the starting and ending reference indices of sub-segment $k$, corresponding to the pre-planned *planning frames*. The hyperparameter $K$ specifies the size of the recent-context window on which the AR decoder conditions—namely, the $K$ frames immediately preceding $s_k$. Because $\{s_k - K : s_k\}$ overlaps with the tail of the previous sub-segment, residual errors inevitably leak into the current one, leading to error propagation across boundaries.

To counteract this bias, we replace the single predecessor at each boundary with a *local multi-frame set*. Concretely,

$$\mathcal{P}_{s_1} = \{2, 3\}, \quad \mathcal{P}_{e_1} = \{10, 11, 12\}, \quad \mathcal{P}_{s_2} \approx \{10, 11, 12\}, \quad \mathcal{P}_{e_2} = \{19, 20\},$$

where $\mathcal{P}_{s_k}$ denotes the local index set of expanded pre-planning frames around boundary $s_k$. Using these expanded anchors, the conditional distribution for sub-segment $k$ is refined to

$$p\big(x^{s_k+1:e_k-1} \mid x^{1:s_k}, x^{e_k}\big) \approx p\big(x^{s_k+1:e_k-1} \mid x^{(s_k-2-K):(s_k-3)}, x^{s_k-2:s_k}, x^{e_k}\big). \tag{6}$$

Conditioning on a compact bundle of early-step, low-drift frames—rather than a single predecessor—dilutes residual errors inherited from the previous sub-segment. At the same time, the model's recency bias naturally prioritizes the most recent elements within this bundle, thereby stabilizing long-horizon synthesis and suppressing cross-boundary error propagation without discarding information from the planned anchors.

## D  ERROR ACCUMULATION ANALYSIS IN AR MODELS

**Autoregressive (AR) Models.** Autoregressive (AR) models generate a sequence $x = (x^1, \ldots, x^T)$ by factorizing its joint probability distribution according to the chain rule of probability:

$$p_\theta(x) = \prod_{t=1}^{T} p_\theta(x^t \mid x^{<t}), \tag{7}$$

where $x^{<t} = (x^1, \ldots, x^{t-1})$ denotes all previously generated elements. In practice, AR models are commonly trained with the *teacher forcing* strategy, which replaces the model's own past predictions with the ground-truth history during training. This reduces the training objective to a standard negative log-likelihood (NLL) minimization:

$$\mathcal{L}(\theta) = -\sum_{t=1}^{T} \log p_\theta(x^t \mid x^{<t}\text{gt}), \tag{8}$$

where $x^{<t}\text{gt}$ denotes the ground-truth prefix of the sequence. Such training ensures stable and efficient optimization, but it also introduces a train-test discrepancy—commonly referred to as *exposure bias* (Ning et al., 2024)—because the model will rely on its own predictions rather than ground-truth history during inference, potentially leading to error accumulation over long sequences.

To analyze the underlying sources and impacts of error accumulation, we follow (Arora et al., 2022) and formulate AR generation as a sequential decision process under the imitation learning (IL) framework. Here, the state is defined as $s^t = x^{<t}$, the action as $a^t = x^t$, the policy as $\pi_\theta(a^t \mid s^t) = p_\theta(x^t \mid x^{<t})$, and the oracle policy as $\pi^*(a^t \mid s^t) = p_{\text{data}}(x^t \mid x^{<t})$. Maximum-likelihood training corresponds to behavior cloning, which minimizes training loss on the oracle-induced state distribution but suffers from compounding errors once the policy is executed on its own rollouts.

In the imitation learning literature (Ross et al., 2011), rolling out a policy trained via behavior cloning often leads to error accumulation. This happens because the policy is executed on its own predictions rather than the oracle states seen during training. To analyze this effect, researchers use inference-time regret, which measures the performance gap between the behavior cloning policy $\pi_{BC}$ and the oracle policy $o$ during rollout:

$$\mathcal{R}(\pi_{BC}) = L^I(\pi_{BC}) - L^I(o). \tag{9}$$

Here, $L^I(\pi)$ denotes the expected cumulative loss (or cost) when executing policy $\pi$ over the entire rollout horizon. Let $\epsilon$ denote the average expected error of executing the behavior cloning policy $\pi_{BC}$ over $T$ steps, which itself is upper-bounded. The regret of behavior cloning is bounded by

$$T\epsilon \leq \mathcal{R}(\pi_{BC}) \leq T^2\epsilon, \tag{10}$$

Building on this analysis, and following (Arora et al., 2022), we further extend it to the AR video generation setting with model $p_\theta$ and decoding strategy $\mathcal{F}$, which yields

$$T\epsilon \leq \mathcal{R}(p_\theta, \mathcal{F}) \leq T^2\epsilon, \tag{11}$$

which demonstrates that even small per-step errors can accumulate linearly in expectation and quadratically in the worst case, thereby explaining the progressive drift and long-horizon degradation observed in autoregressive generation under exposure bias.

# E  COMPUTATION COST ANALYSIS

To better understand the computational cost of MMPL, we analyze its stage-wise latency, multi-GPU efficiency, seeding strategies, and the impact of different $(t_a, t_b, t_c)$ configurations. All experiments are conducted using the full 14B model on H100 GPUs without distillation.

## E.1  STAGE-WISE LATENCY.

We denote the four inference steps illustrated as 1–4 in Figure 11 by $a, b, c, d$ in the following analysis. MMPL inference therefore consists of four steps $a, b, c, d$, with execution times denoted as $T_a, T_b, T_c$, and $T_d$. Here, $T_b$ represents the per-segment Micro Planning cost, while $T_c + T_d$ corresponds to Content Population. Macro Planning progressively copies the terminal planning frames of segment $s$ to initialize segment $s+1$, and its total overhead is the sum of these copy operations plus all per-segment Micro Planning costs. The stage-wise breakdown is listed in Table 9, where "∼" indicates negligible overhead and $L$ denotes the number of 5-second segments.

Table 9: Stage-wise latency of MMPL.

| Stage | Latency (s) |
|---|---|
| $T_a$ | 25 |
| $T_b$ | 109 |
| $T_c$ | 113 |
| $T_d$ | 148 |
| Re-encode/Decode | 1.3 |
| Inter-GPU Transfer | ∼ |
| Macro | $25 + L \times 109$ |

## E.2  MULTI-GPU EFFICIENCY.

We report GPU utilization, peak VRAM, and total inference time for generating 60-second videos under standard MMPL settings using 1, 2, and 4 GPUs (Table 10). Increasing the number of GPUs reduces the overall latency. In our experiments, the four GPU configuration already provides strong throughput, and adding more GPUs does not bring a noticeable additional speedup. For this reason, we report results for one, two, and four GPUs, which sufficiently illustrate the behavior of MMPL under multi GPU settings. In addition, we visualize the GPU utilization patterns during 60-second video generation under different GPU configurations, as shown in Fig. 12, which further illustrates how parallel decoding improves hardware efficiency and reduces idle cycles.

Table 10: Multi-GPU efficiency for 60-second video generation.

| GPUs | Avg Util. (%) | Peak VRAM (GB) | Cost Time (s) |
|---|---|---|---|
| 1 | 99.44 | 68.16 | 4465 |
| 2 | 94.31 | 68.16 | 2354 |
| 4 | 80.69 | 68.16 | 1660 |

## E.3  MID-POINT VS. TERMINAL SEEDING.

We evaluate the memory-throughput trade-off between mid-point and terminal seeding using a two-GPU setup generating 10-second videos (Table 11). Mid-point seeding reduces KV-cache length

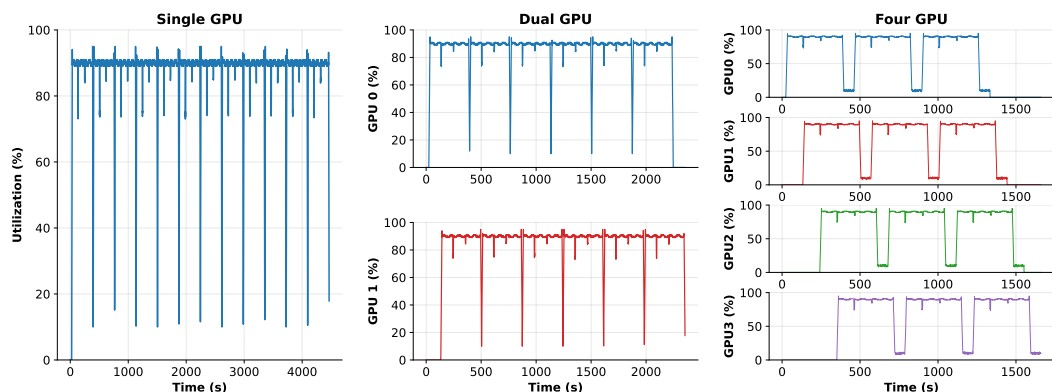

Figure 12: **GPU utilization curves during 60-second video generation.**

and thus memory usage, whereas terminal seeding achieves faster runtime at the cost of higher peak VRAM. This reflects a controllable balance between memory efficiency and throughput.

Table 11: Memory-throughput trade-off of mid-point vs. terminal seeding.

| Strategy | Peak VRAM (GB) | Cost Time (s) |
|---|---|---|
| Mid-point | 55.17 | 586 |
| Terminal | 68.16 | 504 |

### E.4 EFFECT OF $(t_a, t_b, t_c)$ SETTINGS.

We further analyze peak VRAM usage and step-wise latency for various $(t_a, t_b, t_c)$ configurations under the 5-second generation setting (Table 12). The results show that the latency of each stage is jointly influenced by its effective contextual length and the number of frames it is responsible for generating, whereas the peak VRAM is primarily governed by the KV-cache span. In our design, the number of frames whose KV states must be kept in memory scales with the total number of generated frames minus the frames falling between $t_b$ and $t_c$.

Table 12: Peak VRAM and latency for different $(t_a, t_b, t_c)$ configurations.

| $(t_a, t_b, t_c)$ | Peak VRAM | $T_a$ | $T_b$ | $T_c$ | $T_d$ | Total Time |
|---|---|---|---|---|---|---|
| $[2, 3], [10, 11, 12], [19, 20]$ | 68 | 25 | 109 | 113 | 148 | 395s |
| $[2, 3, 4], [10, 11, 12], [18, 19, 20]$ | 70 | 25 | 148 | 101 | 129 | 403s |
| $[2, 3, 4, 5], [10, 11, 12], [17, 18, 19, 20]$ | 72 | 25 | 192 | 88 | 109 | 414s |
| $[2, 3, 4, 5], [11], [17, 18, 19, 20]$ | 70 | 25 | 148 | 101 | 129 | 403s |

## F IMPORTANCE OF VAE

We compare the extrapolation procedure from the public CausVid codebase against our proposed *drift-resilient re-encoding and decoding strategy*, as shown in Figure 13. When extrapolation extends beyond the training context length and requires segment stitching, the baseline exhibits severe color drift and visual artifacts, whereas our method effectively suppresses these degradations. This strategy also mitigates the VAE-induced color drift accumulated across segments. To quantify this effect, we disable the re-encode/decode module and measure boundary quality at the junctions between consecutive 5-second segments. The results in Table 13 show that re-encode/decode substantially improves cross-segment color consistency, as evidenced by significantly better H-channel metrics and a nearly unchanged LPIPS score.

| *Starting frames* | *Extrapolated Frames(Ours)* | *Extrapolated Frames(naive)* |

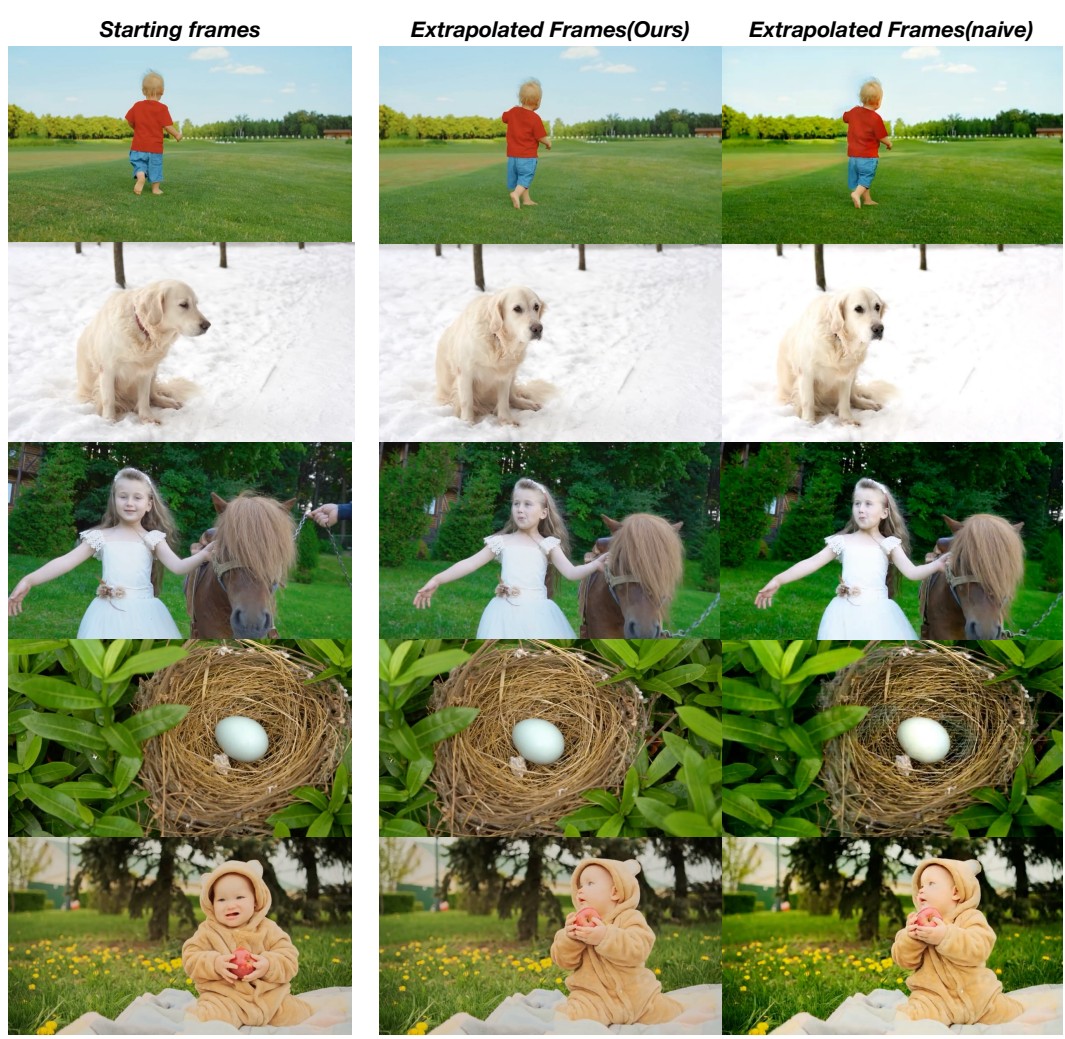

Figure 13: **Qualitative comparisons on video extrapolation.**

Table 13: Boundary quality evaluation across consecutive 5-second segments.

| Variant | H-channel L1 Distance ↓ | H-channel Correlation ↑ | t-LPIPS ↓ |
|---|---|---|---|
| w/o Re-encode/Decode | 0.697 | 0.675 | 0.059 |
| MMPL | **0.028** | **0.999** | **0.023** |

## G    NOISE INITIALIZATION STRATEGY FOR SMOOTHING GENERATION

In this work, we propose a specialized noise initialization strategy to address potential temporal discontinuities and instability at the transition boundaries between planning frames and content frames as shown in Figure 14. This approach ensures smooth visual transitions by strategically incorporating noise information from adjacent planning frames during the content frame generation process. Let $P_{n-1}$ and $P_n$ represent the planning frames at temporal positions $n-1$ and $n$, respectively, and let $C_{n+1}$ denote the target content frame at position $n+1$. To establish the theoretical foundation, we first recall the standard diffusion forward process formulation. Given a clean frame $\mathbf{x}_0$ at diffusion timestep $t$, the noisy observation $\mathbf{x}_t$ is generated through the Gaussian perturbation:

$$q(\mathbf{x}_t|\mathbf{x}_0) = \mathcal{N}(\mathbf{x}_t; \sqrt{\bar{\alpha}t}\mathbf{x}_0, (1 - \bar{\alpha}t)\mathbf{I}), \tag{12}$$

where $\bar{\alpha}_t$ denotes the cumulative product of the noise schedule coefficients. This process can be equivalently expressed as:

$$\mathbf{x}_t = \sqrt{\bar{\alpha}t} \cdot \mathbf{x}_0 + \sqrt{1 - \bar{\alpha}t} \cdot \epsilon, \quad \epsilon \sim \mathcal{N}(\mathbf{0}, \mathbf{I}). \qquad (13)$$

Building upon this formulation, our methodology initializes the noise vector $\epsilon_{C_{n+1}}$ for the content frame $C_{n+1}$ through a weighted interpolation of the noise vectors associated with the preceding planning frames. Specifically, the initialization follows:

$$\epsilon_{C_{n+1}} = \alpha \cdot \epsilon_{P_n} + (1 - \alpha) \cdot \epsilon_{P_{n-1}}, \qquad (14)$$

where $\epsilon_{C_{n+1}}$ represents the noise vector utilized in the reverse diffusion process for generating content frame $C_{n+1}$, $\epsilon_{P_n}$ and $\epsilon_{P_{n-1}}$ correspond to the noise vectors derived from planning frames $P_n$ and $P_{n-1}$ This noise initialization strategy ensures a continuous evolution of stochastic patterns across frame boundaries, effectively mitigating visual artifacts and temporal inconsistencies. By controlling the interpolation weight $\alpha$, our method provides precise adjustment over the temporal smoothness characteristics, enabling stable and coherent video generation while maintaining high visual quality throughout the sequence. To verify its contribution, we disable the noise-interpolation module and evaluate boundary quality specifically at these boundaries. These results in Tab.14 show that noise interpolation substantially improves the stability of segment transitions and effectively eliminates content jumps, as evidenced by nearly identical H-channel metrics and a significantly reduced LPIPS score.

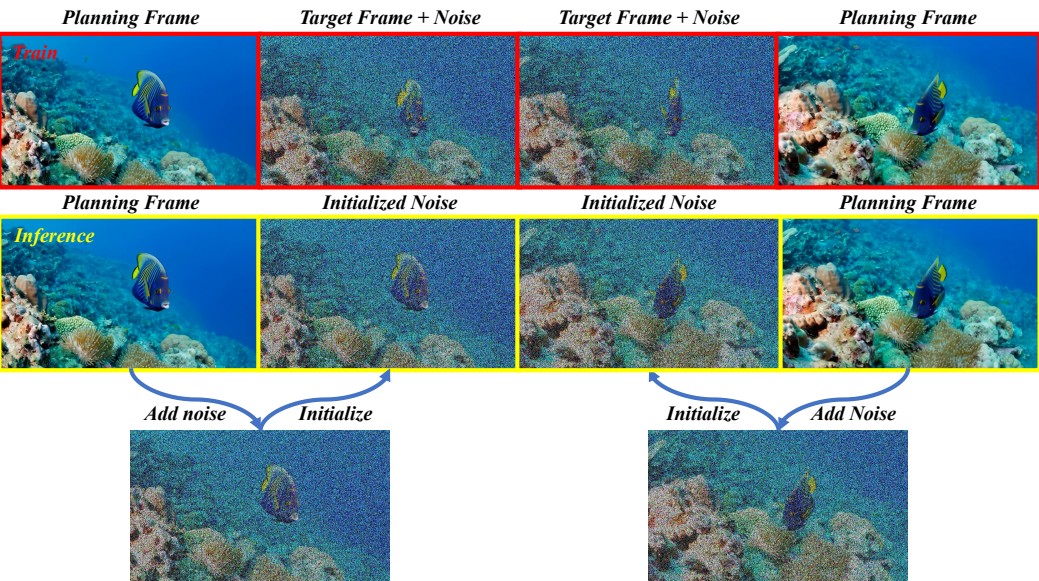

Figure 14: Framework for stable and smoothing frame generation via coherent noise initialization.

Table 14: Boundary quality evaluation at planning-to-content transition points.

| Variant | H-channel L1 Distance ↓ | H-channel Correlation ↑ | t-LPIPS ↓ |
|---|---|---|---|
| w/o Noise-Interpolation | 0.187 | 0.976 | 0.416 |
| MMPL | **0.051** | **0.998** | **0.030** |

## H  FAILURE CASES

We also examine typical failure cases that occur when key stabilization components are removed. Without coherent noise initialization or the re-encode/decode step, adjacent segments may no longer share consistent appearance statistics, leading to noticeable color shifts and abrupt content changes at segment boundaries. These effects are illustrated in Figure 15, where the absence of these components results in visible boundary inconsistencies, while the full MMPL setup maintains smooth and stable transitions.

*Color Shift*               *Content flicker*

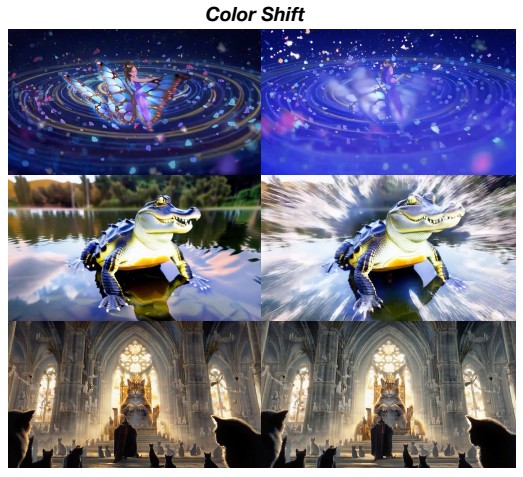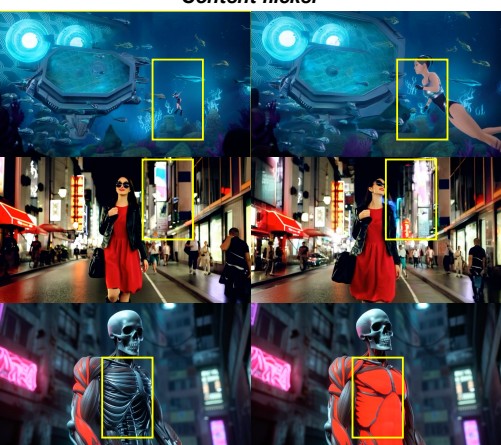

Figure 15: Failure cases for long video generation.

# I FULL VBENCH BENCHMARK EVALUATION

We conduct a comprehensive evaluation on the full VBench benchmark rather than VBench-long using 30-second video clips, covering all 946 prompts and all 16 metrics reported in Tables 15 and 16. All scores are obtained using the official standardized evaluation scripts, and we refer readers to the VBench documentation for detailed metric definitions. Overall, MMPL achieves the highest Quality Score, demonstrating clear advantages in long-video generation. At the same time, MMPL exhibits competitive Dynamic Degree performance. We regard this as a desirable balance: the planning mechanism significantly enhances long-range temporal stability and perceptual coherence while still preserving adequate short-term motion variation, resulting in a well-calibrated trade-off between dynamic expressiveness and long-horizon consistency.

Table 15: Comparison on VBench quality metrics for 30-second videos. [‡] Indicates results reported by related work.

| Model | Subject Consistency | Background Consistency | Temporal Flickering | Motion Smoothness | Dynamic Degree | Aesthetic Quality | Imaging Quality | Quality Score |
|---|---|---|---|---|---|---|---|---|
| CausVid[‡] | 89.50 | 90.00 | 99.41 | 98.06 | 63.88 | 61.82 | 65.30 | 80.89 |
| Self Forcing[‡] | 88.61 | 89.53 | 98.90 | 98.57 | 68.05 | 60.60 | 68.98 | 81.39 |
| **MMPL (ours)** | **92.26** | **94.16** | 99.11 | **98.83** | 61.11 | **62.77** | 65.27 | **82.47** |

Table 16: Comparison on VBench semantic metrics for 30-second videos. [‡] Indicates results reported by related work.

| Model | Object Class | Multiple Objects | Human Action | Color | Spatial Relationship | Scene | Temporal Style | Appearance Style | Overall Consistency | Semantic Score |
|---|---|---|---|---|---|---|---|---|---|---|
| CausVid[‡] | 78.56 | 58.84 | 81.00 | 81.02 | 59.62 | 31.32 | 22.51 | 20.04 | 23.16 | 65.85 |
| Self Forcing[‡] | 80.06 | 62.88 | 83.00 | 79.80 | 74.76 | 30.59 | 23.78 | 20.41 | 24.80 | 69.17 |
| **MMPL (ours)** | 78.25 | 57.24 | 80.00 | **82.46** | 73.84 | 29.91 | **24.34** | 19.76 | 24.40 | 67.91 |

# J DISCLOSURE OF LARGE LANGUAGE MODEL (LLM) USAGE

In this paper, we used Large Language Models (LLMs) to assist in various aspects of the writing process. Specifically, LLMs were employed to help polish the writing, improve clarity, and enhance the overall presentation of the text. The models were utilized to provide suggestions for improving the grammar, coherence, and flow of certain sections of the manuscript. This assistance was integral to the refinement of the paper's language, but all scientific content, methodology, and conclusions were independently developed by the authors. The use of LLMs is limited to language-related tasks and does not extend to the intellectual contributions to the research findings or data analysis.

## K SCALABILITY

### K.1 IMAGE-TO-VIDEO EXTENSION

Our framework is not restricted to the text-to-video (T2V) task; it can be seamlessly extended to image-to-video (I2V) generation without introducing any architectural modifications or additional image encoders. This flexibility derives from the unified autoregressive design, which only requires lightweight adjustments to the number and ordering of autoregressive steps. As a result, the framework adapts naturally to different input modalities while maintaining temporal consistency and generation quality as shown in Figure 16.

*Reference Image*                          *Generated Video*

Figure 16: **Qualitative results of extending our unified autoregressive framework from text-to-video (T2V) to image-to-video (I2V) generation.** Without any architectural modifications or additional image encoders, the framework adapts seamlessly by only adjusting the number and ordering of autoregressive steps, while preserving temporal consistency and visual quality.

## K.2 Adaptation to Self-Forcing and DMD

Our approach can be seamlessly integrated with self-forcing strategies without any architectural modifications. Specifically, it only requires adjusting the attention visibility range and the prediction order during both training and inference. This lightweight adaptation enables direct compatibility with existing self-forcing pipelines, while retaining the benefits of our planning-based design. Combined with parallelized decoding, the resulting system achieves substantial inference speedups, sustaining over 32 FPS in long-horizon video generation as shown in Figure 17.

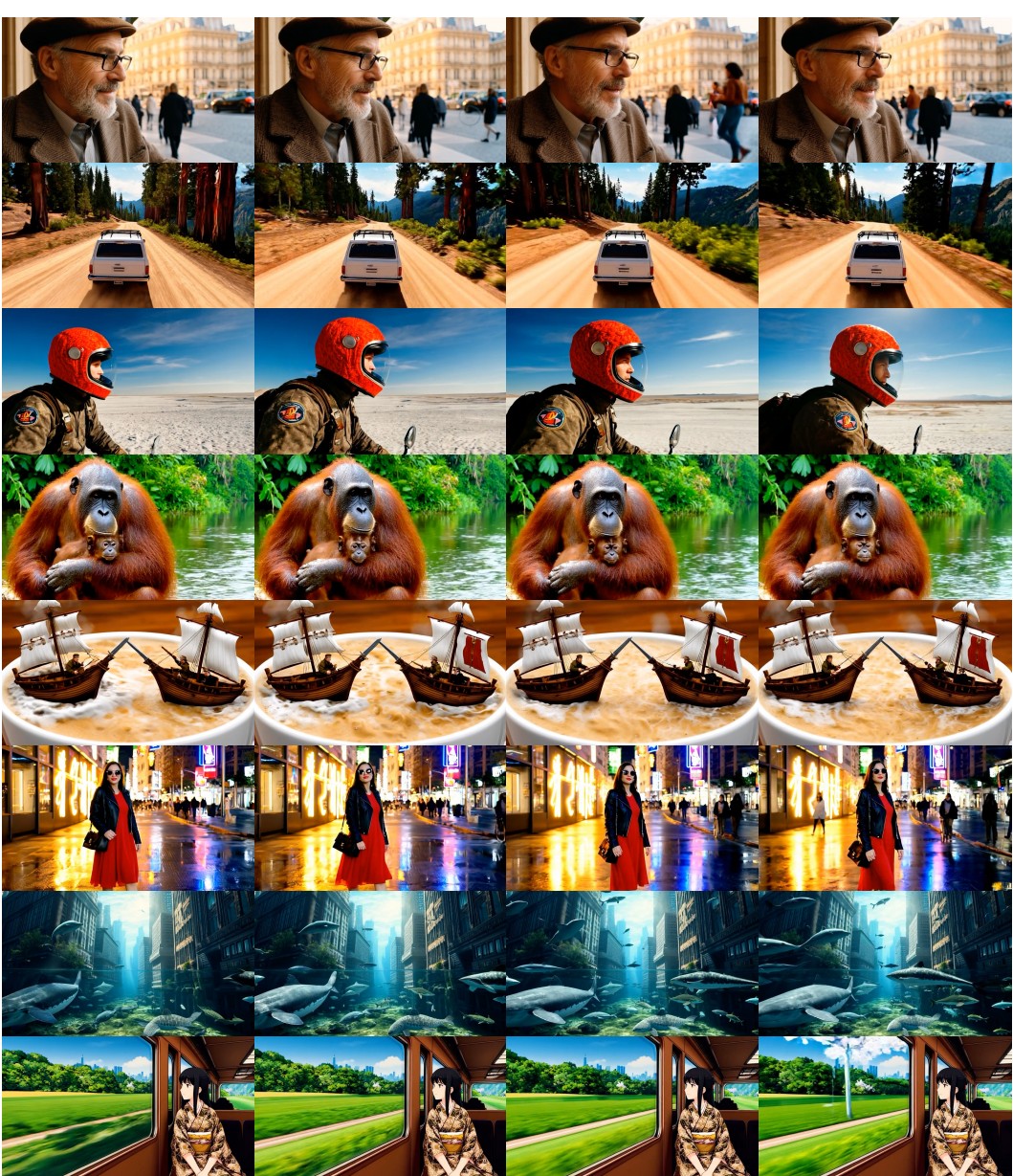

Figure 17: **Integration of our framework with Self Forcing (Huang et al., 2025a) and DMD (Yin et al., 2025) strategies**. The adaptation requires no architectural changes—only modifications to the attention visibility range and prediction order during training and inference. Combined with parallelized decoding, the method achieves substantial inference acceleration, sustaining over 32 FPS in long-horizon video generation.

## L    MORE QUALITATIVE RESULTS

To better demonstrate the robustness of our model, we present additional experimental results on 30s long video generation, as shown in Figure 18.

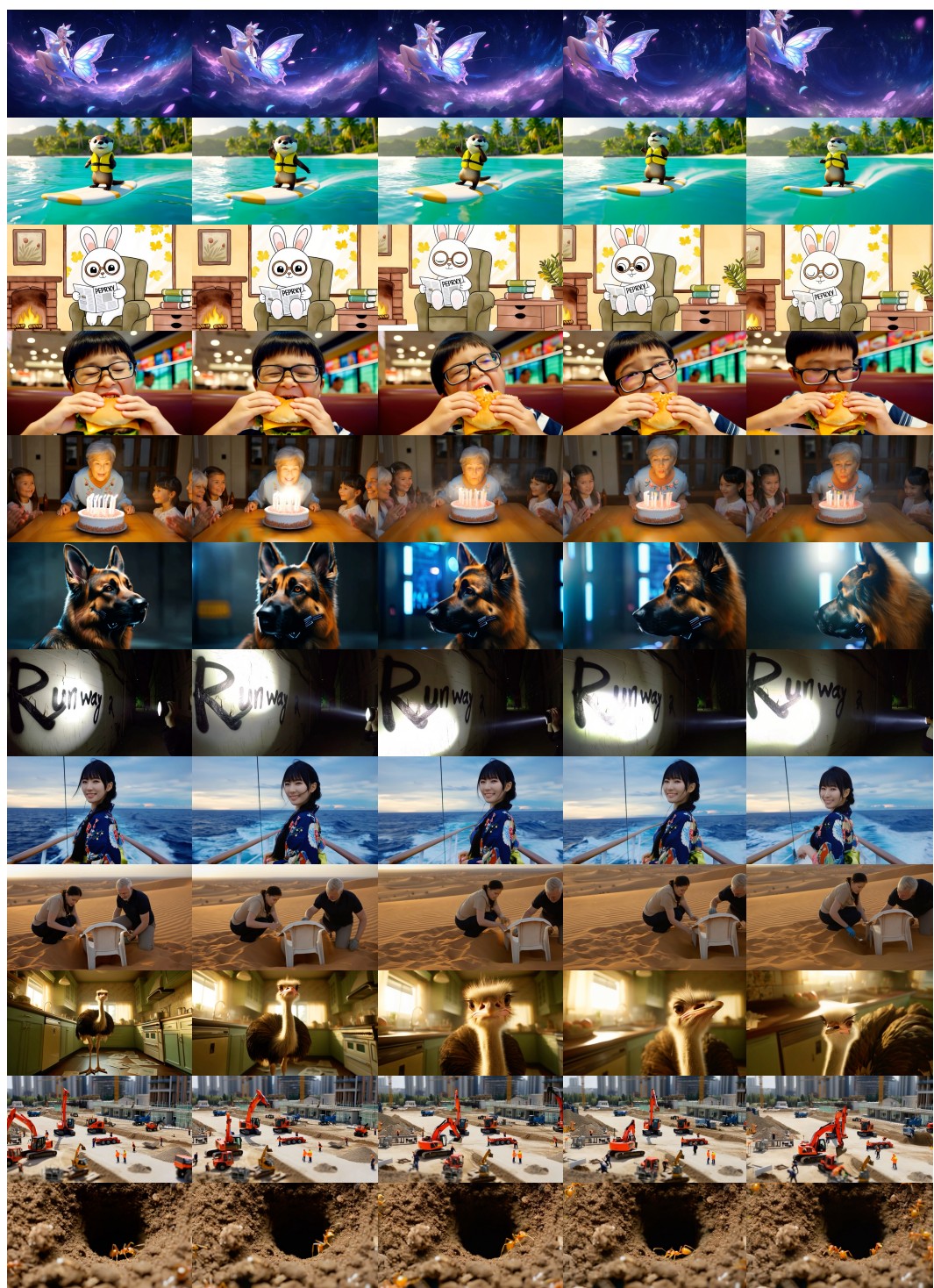

Figure 18: **Additional qualitative results of 30s long video generation.** Our model produces temporally coherent and visually consistent sequences across diverse scenarios, further demonstrating its robustness.

