# OpenReview forum: "Macro-from-Micro Planning for High-Quality and Parallelized Autoregressive Long Video Generation"
_ICLR.cc/2026/Conference — Submitted to ICLR 2026_

### Official Review · Reviewer_R1mP · 2025-10-26

**Soundness:** 2
**Presentation:** 1
**Contribution:** 2
**Rating:** 4
**Confidence:** 4

**Summary:**

The paper proposes a macro-micro planning strategy for long video generation. The micro planning phase initially predicts a sparse set of pre-planning frames, which represent the early, mid, and terminal frames of each segment. Then, the macro planning phase predicts the intermediate frames within each segment, conditioned on the predicted first frames from the micro planning. This hierarchical setup can be further parallelised, enabling the natural utilisation of a multi-gpu setup for further latency reduction. The proposed network achieves comparable performance to other inference optimisation methods such as Self-forcing and CausVid on VBench, and demonstrates significantly superior preference rate with human evaluation.

**Strengths:**

The proposed method is simple and easy to implement. It’s an independent strategy that can be easily combined with self-forcing and other inference speed-up techniques.

**Weaknesses:**

- The linked anonymous demo page cannot be opened.
- The overall paper writing is poor, with many inconsistent notations and visualisations.
    - Figure 2’s caption only describes the limitations of current AR methods, without mentioning the proposed method.
    - Figure 3 also raises questions. Should micro-planning predict frames across all segments, or just segment 1? Similarly, macro-planning should jointly optimise for three frames rather than autoregressively as indicated in the figure caption.
    - In Section 3.1, M is defined as the Micro planning network or process, but it is also defined as a set containing three predicted frames. This contradicts the example provided in Appendix Figure 11, where the planning frames for early, midpoint, and terminal can be more than one frame.
    - Similarly, M+ is defined as the macro-planning network or processing, but it is also defined as the number of predicted frames.
    - Equation 2 is incorrect because the conditioning generative process conditioned on x^1 is not included in the expanded formula.
    - In Figure 6, t_i represents an timestep or an inference step?
- The proposed method is built on top of WAN 2.1 14B, while the important baselines, CausVid and Self-forcing, are both distilled to 1.3B models. This makes it unfair to compare them directly. Is it possible to directly report results on the 1.3B model as well, and ensure that all models generate the same length of output?

**Questions:**

- L192: earyl → early
- L355: t_a=2,3,4 is inconsistent with Fig 11 t_a=2,3.

---

> ### Author Response · Authors · 2025-11-26
>
> Dear reviewer R1mP, we sincerely thank for your constructive feedback, insightful questions, and valuable suggestions. Below, please find our responses to your concerns.
>
> > **Weakness 1**: Anonymous Demo Page.
>
> We apologize for the inconvenience. Our demo is hosted on Anonymous GitHub, which appears to have experienced a temporary service interruption during the review period. This was an issue on the hosting side, not ours. The link is now fully functional and has been independently verified to work without issues. In addition to the original demo page, we also provide a direct anonymous link to all video examples to avoid similar issues in the future: https://anonymous.4open.science/r/Anonymous_MMPL_Webpage_Submission_Number_456/
>
> > **Weakness 2, Question 1 and Question 2**: Presentation Issues.
>
> Thank you for this helpful comment. It has greatly guided us in improving the readability and overall presentation of the paper. In the revised version **(Section 3 and Section 4)**, we have unified all notations, removed ambiguous symbol reuse, and refined all figures for consistent and clear presentation. We would also like to clarify a few points in response to your comment.
>
> 1. Figure 2 illustrates only AR-based methods and does not depict our approach. It is included in the Related Work section to highlight limitations of existing baselines. We have updated the caption in **Section 2** to make this purpose explicit.
> 2. Figure 3 is consistent with our method description. The micro-planning module predicts only the planning frames within its own segment and does not operate across segments. The macro-planning module takes the planning frames from the previous segment as input and predicts the planning frames for the current segment (i.e., its micro-planning targets). We have revised **Figure 3** and its caption to make this workflow clearer.
> 3. Due to space limitations, some computational details were condensed in the original submission. Moreover, in the revised version, we add clearer explanations and efficiency analyses in **Section E** to better help the reviewer understand the additional details.
>
> > **Weakness 3**:  Fairness of Experiments.
>
> Thank you for raising this important point. To ensure fairness, we use the same backbone and match video lengths across all methods. We evaluate performance using VBench-long quality metrics and an additional drift metric. Following Reviewer b7Dw’s suggestion, this drift metric computes the H-channel L1 distance and correlation between the first and last frames of each 30-second video, providing a direct measure of long-range color drift.
>
> **For fairness, all 1.3B models are trained with distillation from the 14B teacher, while the 14B model itself does not use any distillation.**
>
> 1. **30s long video generation performance on 1.3B models:**
>
>     | Model        | Params |  Subj  | Back. |  Mot. |  Aes. | Img. | H-channel L1| H-channel Corr|
>     |:-------------|:------:|:------:|:-----:|:-----:|:-----:|:----:|:---------------:|:----------------:|
>     | CausVid      | 1.3B   | 0.969  | 0.980 | 0.981 | 0.606 | 0.661 |     0.711       |      0.598       |
>     | Self-Forcing | 1.3B   | 0.967  | 0.958 | 0.980 | 0.593 | 0.689 |     1.152       |      0.162       |
>     | **MMPL-1.3B**   | 1.3B   | **0.980** | 0.970 | **0.987** | 0.600 | 0.665 |   **0.348**     |    **0.923**     |
>
>
> 2. **30s long video generation performance on large-scale models:**
>
>     | Model        | Params | Subj. | Back. | Mot. |  Aes.  |  Img.  | H-channel L1| H-channel Corr|
>     |:-------------|:------:|:-----:|:-----:|:----:|:------:|:------:|:---------------:|:----------------:|
>     | Skyreels-v2  | 14B    | 0.956 | 0.966 | 0.991 | 0.600  | 0.581  |       0.902     |      0.503       |
>     | **MMPL-14B**   | 14B    | **0.980** | **0.968** | **0.992** | **0.628** | **0.661** |   **0.306**     |    **0.927**     |
>
> These results confirm that the comparison is fair: under the same backbone, same video length, and same distillation settings, MMPL consistently achieves higher quality and much lower color drift. This shows that the improvements come from the MMPL design itself, rather than from training differences.

---

### Official Review · Reviewer_b7Dw · 2025-10-30

**Soundness:** 2
**Presentation:** 3
**Contribution:** 2
**Rating:** 6
**Confidence:** 4

**Summary:**

The paper proposes Macro-from-Micro Planning (MMPL) for long video generation with autoregressive (AR) diffusion backbones. The core idea is to (1) plan sparse keyframes per segment (Micro Planning) from the first frame of a segment, (2) chain these micro plans across segments (Macro Planning) by seeding the next segment with the previous segment’s planned endpoints, and then (3) populate all in-between frames in parallel across segments, enabled by the fixed planning anchors. To further increase throughput on multi-GPU setups, the authors add an adaptive workload scheduler with two modes that trade memory peak vs. throughput by choosing whether the next segment starts from the mid-point or terminal planned frame of the previous segment. The method is implemented by fine-tuning Wan2.1-T2V-14B on a curated ~50k video set with LLM-generated captions, and evaluated on VBench-long plus a small user study on 19 MovieGen prompts. Reported results show higher VBench scores and strong user preference, as well as ~3× wall-clock speedups with 4 GPUs for 60-second videos under their pipeline.

**Strengths:**

1. Making long-video AR generation as plan-then-populate with keyframes per segment, then parallel population conditioned on those anchors, is a clean, modular perspective. While planning and hierarchical generation exist, the specific coupling of joint keyframe prediction from the segment’s first frame plus segment-level AR chaining is crisply articulated and differs from step-jump or packed-context approaches.
2. The two-mode scheduler (minimum memory peak vs. maximum throughput) provides an explicit, practical deployment knob that many papers hand-wave. The interplay between when to start the next segment and how to reuse/skip frames is well described.
3. The error-accumulation analysis connects AR drift to exposure bias and imitation learning regret bounds, motivating the reduction from frame-wise to segment-wise dependency depth under MMPL. While mostly qualitative/upper-bound reasoning, it grounds the need for planning anchors and shorter causal chains.
4. The stability tooling around re-encode/decode across segment boundaries and an appendix noise-initialization scheme for smoother transitions addresses practical artifacts (color shift/flicker) that often undermine long-video demos.

**Weaknesses:**

1. Questionable novelty boundary vs. prior “planning/parallel AR” lines. The closest prior they cite, FramePack-Plan, already reduces error via step-wise frame jumping and context compression; other works use hierarchical/story planning or parallelized AR decoding. The paper states three innovations (two-level plan, one-pass segment keyframe prediction, then parallel population), but the empirical study does not isolate what is fundamentally new versus combinations/engineering of known techniques. A finer related-work positioning and head-to-head comparisons against frame-jump/packed-context methods are missing.
2. The macro vs. micro contribution is not ablated: we see ablations for positioning of planning frames and training policies, but not for “Micro only”, “Macro only”, “Population parallelism only”, or “Scheduler off”. This makes it hard to credit the gains to the planning hierarchy versus downstream parallel population and scheduling.
4. implementations and backbones vary (e.g., Wan-14B fine-tuned for the proposed method, distilled 1.3B baselines among others). The paper says “comparable scale,” but the exact compute/training budgets and data curation (LLM captions; selection of top 1% licensed clips by LAION aesthetic scoring) could favor the proposed system’s data quality. A deeper fairness audit (same base model, same finetuning budget) is needed.
5. The choice of mid-point vs. terminal seeding trades memory vs. throughput, yet we lack measured curves (utilization, peak VRAM, effective FPS) to guide practitioners.
6. Missing or thin ablations on stability modules. The re-encode/decode stitching and noise-interpolation initialization (Appx.) look helpful, but there are no quantitative ablations isolating their impact (e.g., color-shift metrics, boundary flicker metrics, LPIPS/temporal warping vs. off). Without this, it’s hard to know how much they contribute versus planning alone.

**Questions:**

1. Can you provide direct comparisons to FramePack-Plan (and other planning/packed-context or parallel-AR methods) under the same backbone, training budget, and prompts? Ideally include Micro-only, Macro-only, Parallel-only, and Scheduler-off ablations to quantify the unique value of each component.
2. Please include stage-wise wall-clock breakdowns (Micro Plan, Macro chain latency, Content Population, re-encode/decode, inter-GPU transfer) and utilization curves for 1/2/4/8 GPUs, with peak VRAM and throughput across different ta/tb/tc settings and segment sizes. How sensitive is the speedup to segment granularity and anchor spacing?
3. You mention inevitable prefix delays before parallelism ramps. Can you quantify this overhead and propose a look-ahead or speculative planning scheme to reduce it? Do you consider asynchronous seeding (e.g., starting s+2 from xtb of s before s+1 finishes)?
4. Please add ablations toggling re-encode/decode and the noise-interpolation initialization (Fig. 13) with objective boundary stability metrics (e.g., temporal color histograms near joins, t-LPIPS at boundaries). Also report any failure cases (banding, ghosting).

---

> ### Author Response · Authors · 2025-11-26
>
> Dear reviewer b7Dw, we sincerely thank for your constructive feedback, insightful questions, and valuable suggestions. Below, please find our responses to your concerns.
>
> > **Weakness 1**: Novelty Boundary Between other Methods.
>
> Thank you for raising this important point. To clarify the novelty boundary, we discuss MMPL’s distinctions from both planning-based methods and parallel AR approaches.
>
> 1. **Comparison with Other Planning Methods:**
> MMPL and FramePack-Plan are concurrent works developed independently. To maintain scholarly rigor, we outline their conceptual and technical differences below.
>
>     - **Approach:** MMPL aims at long-range temporal planning for the entire video storyline, whereas FramePack’s planning is limited to the current local segment only.
>
>     - **Efficiency:** MMPL enables parallel multi-segment generation and mitigates long-range error accumulation, which FramePack cannot inherently achieve.
>
>     - **Scalability:** MMPL integrates seamlessly with DMD method, supporting near–real-time long-video generation. FramePack-Plan does not provide such real-time capabilities.
>
> 2. **Comparison with Other Parallel AR Methods:**
> To the best of our knowledge, existing AR-based video generation models remain constrained by the chain rule of autoregression and do not support multi-GPU or multi-node parallel prediction. Our work fundamentally breaks this constraint, introducing a novel, parallelizable AR framework that represents a key novelty in the field.
>
> 3. **Fundamental Novelty Beyond Engineering Combinations:**
> MMPL is fundamentally new because it breaks the standard autoregressive chain rule through a global long-range planning mechanism that enables parallel multi-segment generation and reduces long-term error accumulation. This is a new formulation, not an engineering combination of existing methods.

---

> > ### Author Response · Authors · 2025-11-26
> >
> > > **Weakness 2, Weakness 3 and Question 1**: Experiments and Ablations under same settings.
> >
> > Thank you for your suggestion. We further evaluate performance using VBench-long quality metrics and the drift metric introduced in JEPA [1], which captures temporal consistency over long videos.
> >
> > *[1] Gaussian Embeddings: How JEPAs Secretly Learn Your Data Density, arXiv, 2025.*
> > > JEPA is originally designed to estimate dataset-level distributions, and we adopt a simple reinterpretation: treating a single video as a dataset and each frame as a sample. Under this view, taking the first frame as the distributional center, the JEPA standard deviation naturally reflects the amount of temporal drift throughout the video.
> > (JEPA-Score Std ↓ = more stable; First–Last Frame Diff ↓ = more consistent)
> >
> >
> > 1. **Comparison using the same backbones:**
> > To ensure fairness and highlight long-video advantages, we evaluate all methods on 30-second videos using the same backbones, same base model and same budget.
> >
> >     | Model        | Params | Subj. | Back. | Mot. | Aes. | Img. | JEPA-Score Std | First–Last Frame JEPA-Score Diff |
> >     |:-------------|:------:|:-----:|:-----:|:----:|:----:|:----:|:--------------:|:--------------------------------:|
> >     | CausVid      | 1.3B   | 0.969 | 0.980 | 0.981 | 0.606 | 0.661 |    0.1107       |              0.4093               |
> >     | Self-Forcing | 1.3B   | 0.967 | 0.958 | 0.980 | 0.593 | 0.689 |    0.2293       |              1.0695               |
> >     | **MMPL-1.3B**   | 1.3B   | **0.980** | 0.970 | **0.987** | 0.600 | 0.665 |    **0.0955**       |              **0.0531**
> >
> > 2. **Comparison with Other Planning Methods:**
> > To ensure fairness, we compare MMPL with planning-based baselines under matched model scales. Since FramePack-Plan is not open-sourced and the released FramePack code only supports image-to-video inference, we adopt a practical alternative: we feed FramePack with the first frame generated by our method.
> >
> >     | Model        | Params | Subj. | Back. | Mot. | Aes. | Img. | JEPA-Score Std | First–Last Frame JEPA-Score Diff |
> >     |:-------------|:------:|:-----:|:-----:|:----:|:----:|:----:|:----------------:|:--------------------------------:|
> >     | FramePack    | 13B    | 0.987 | 0.971 | 0.996 | 0.607 | 0.638 |     0.0853        |               0.1364              |
> >     | **MMPL-14B**   | 14B    | 0.980 | 0.968 | 0.992 | **0.628** | **0.661** |     **0.0705**        |               **0.0281**              |
> >
> > 3. **Ablation on Each MMPL Component:**
> > To quantify the contribution of each component, we evaluate **Micro-only, Macro-only and Scheduler-off variants**. In the Micro-only setting, each segment is generated independently without global planning. In the Macro-only setting, the entire 30-second video is solely composed of the planning frames. A **Parallel-only variant** is not included, because parallel decoding itself relies on the planning module to define segment boundaries and provide the necessary guidance; therefore, it cannot function as an independent ablation.
> >
> >     | Model          | Subj. | Back. | Mot.  | Aes.  | Img.  |
> >     |:---------------|:-----:|:-----:|:-----:|:-----:|:-----:|
> >     | Micro-only     | 0.958 | 0.954 | 0.981 | 0.635 | 0.675 |
> >     | Macro-only     | 0.970 | 0.963 | 0.976 | 0.615 | 0.645 |
> >     | Scheduler-off  | 0.978 | 0.965 | 0.991 | 0.631 | 0.658 |
> >     | **MMPL**       | **0.980** | **0.968** | **0.992** | 0.628 | 0.661 |
> >
> >     Our full method achieves the strongest temporal stability and long-range consistency. In comparison, Micro-only shows high single-frame quality because segments are generated independently; as it has no cross-segment linkage, it avoids error accumulation, which we believe explains its strong per-frame scores despite weak long-range coherence.

---

> ### Author Response · Authors · 2025-11-26
>
> > **Weakness 4, Question 2**: Stage-wise Latency and Multi-GPU Efficiency Analysis.
>
> To facilitate the discussion, we **first provide** several key definitions and underlying principles.
>
> > 1.  We use $(a,b,c,d\)$ to denote the four inference steps in Figure 11, with $T_a\$, $T_b\$, $T_c\$, and $T_d\$ representing their execution times. Here, $T_b\$ is the per-segment micro-planning cost, and $(T_c+T_d\)$ is the Content Population cost. Macro planning iteratively copies the terminal planning frames of segment $i\$ to initialize segment $(i+1\)$. so its overhead can be viewed as the sum of all copy operations plus the micro-planning costs of all segments.
> > 2. Peak VRAM is mainly dominated by the KV-cache length, which equals the total number of generated frames *excluding* those in the final inference step. Under this formulation, the KV-cache size becomes $\text{len}(a{+}b{+}c)$ for the mid-point scheme and $\text{len}(a{+}b{+}c{+}d)$ for the terminal scheme.
> > 3. All tests conducted on a single H100 GPU using the full 14B model (without distillation).
>
> 1. **Stage-wise Latency:**
> We provide the stage-wise wall-clock breakdowns as follows. Here, “~” indicates negligible latency, and $L\$ denotes the number of 5-second segments.
>
>     | Stage          | Latency / (s) |
>     |:---------------:|:-------------:|
>     | $T_a\$            |     25      |
>     | $T_b\$            |     109     |
>     | $T_c\$            |     113     |
>     | $T_d\$           |     148     |
>     | Re-encode/decode   |      1.3      |
>     | Inter-GPU transfer |       ~       |
>     | Macro              |     $25 + L \times 109$      |
>
> 2. **Multi-GPU Efficiency Analysis under Standard Settings:**
> We report GPU-utilization curves for 1, 2, and 4 GPUs, together with peak VRAM usage and throughput under the standard MMPL inference settings for generating 60-second videos. These curves will be included in the revised manuscript in **Section E.3**. We report results up to 4 GPUs, which already achieve near-optimal parallel efficiency for MMPL. At this scale, the pipeline saturates its intended concurrency, and larger GPU counts do not meaningfully improve the end-to-end latency. Therefore, 4 GPUs represent a practical and efficient configuration for MMPL in typical deployment settings.
>
>
>     | #GPUs | Average GPU Utilization (%) | Peak VRAM (GB) | Cost Time (s) |
>     |:-----:|:--------------------:|:--------------:|:----------------:|
>     |   1   |         99.44          |     68.16     |      4465            |
>     |   2   |         94.31          |     68.16     |      2354            |
>     |   4   |         80.69          |     68.16     |      1660            |

---

> ### Author Response · Authors · 2025-11-26
>
> 3. **The choice of mid-point vs. terminal seeding trades memory vs. throughput:**
> We illustrate the mid-point and terminal seeding schemes using a **two-GPU setup** generating an **10-second** video.
>
>     | Strategy   | Peak VRAM (GB) | Cost Time (s)  |
>     |:-----------|:--------------:|:--------------:|
>     | Mid-point  |     55.17      |     586        |
>     | Terminal   |     68.16      |     504        |
>
> 4. **Peak VRAM and throughput across different ta/tb/tc settings and segment sizes:**
> We further report peak VRAM usage and throughput across different $(t_a, t_b, t_c)$ settings (mentioned in **Section 3.1**).
>
>     > Here, $t_a$, $t_b$, and $t_c$ denote the frame indices that define the boundaries of the four inference stages in Section C.2, and $T_a$, $T_b$, $T_c$, and $T_d$ represent the corresponding execution times of each stage. For example, in the first row $[2,3]$, $[10,11,12]$, $[19,20]$, the first stage predicts frames $0$ and $1$; the second stage predicts frames $2$, $3$, $10$, $11$, $12$, $19$, and $20$; the third stage predicts frames $4$, $5$, $6$, $7$, $8$, and $9$; and the final content-population stage predicts the remaining frames.
>
>
>     | \(t_a, t_b, t_c\)                         | Peak VRAM (GB) |$T_a\$ | $T_b\$| $T_c\$| $T_c\$ | Cost Time (s) |
>     |:------------------------------------------|:--------------:|:------:|:------:|:------:|:------:|:--------------:|
>     | [2,3], [10,11,12], [19,20]          |     68      |   25   |   109  |  113   |   148  |     395s       |
>     | [2,3,4], [10,11,12], [18,19,20]     |     70      |   25   |   148  |  101   |   129  |     403s       |
>     | [2,3,4,5], [10,11,12], [17,18,19,20]|     72      |   25   |   192  |   88   |   109  |     414s       |
>     | [2,3,4,5], [11], [17,18,19,20]      |     70      |   25   |   148  |  101   |   129  |     403s       |
>
>     The results show that the latency of each stage is jointly influenced by its effective contextual length and the number of frames it is responsible for generating, whereas the peak VRAM is primarily governed by the KV-cache span. In our design, the number of frames whose KV states must be kept in memory scales with the total number of generated frames minus the frames falling between $t_b$ and $t_c$.

---

> ### Author Response · Authors · 2025-11-26
>
> > **Question 3**: Prefix Delays and Look-ahead Scheme.
>
> 1. **Prefix Overhead:**
> Thank you for the insightful suggestion. To quantify the prefix overhead, we compare two settings using a 4-GPU setup for 60-second videos. Following Section 3.3, the Base setting must wait until all planning frames for every segment are generated before parallel decoding can begin. In contrast, MMPL can already start generating segment $s+2$ from the intermediate state $x^{t_b}_{s+1}$ before segment $s+1$ finishes. This early-start mechanism significantly reduces the prefix overhead and leads to higher overall throughput.
>
>     | Strategy   | Peak VRAM (GB) | Time (s) |
>     |:-----------|:--------------:|:--------------:|
>     | Base     |      68.16          |      2116          |
>     | MMPL     |      68.16          |      1660          |
>
> 2. **Look-ahead Planning Scheme:**
> Regarding the reduction of prefix delays, we already explore look-ahead planning and speculative parallelization in Section 3.3. As you noted, starting segment $s+2$ from the intermediate state $x^{t_b}_{s+1}$ before segment $s+1$ finishes directly corresponds to our method. We appreciate the suggestion, and it is encouraging that your insight aligns with our parallel execution design.
>
>
> > **Question 4**: Additional Ablations Results on toggling re-encode/decode and the noise-interpolation initialization (Fig. 13) with objective boundary stability metrics.
>
> Thank you for the valuable suggestion and for highlighting perspectives beyond standard VBench metrics. In response, we ablate the re-encode/decode step and the noise-interpolation initialization, and evaluate boundary stability using temporal color-histogram consistency and t-LPIPS. Representative failure cases are also included in the revised manuscript in **Section H**.
>
> 1. **Re-encode/Decode:**
> This strategy mitigates color drift caused by the VAE. We disable it and evaluate boundary quality at the junctions between consecutive 5-second segments:
>
>     | Variant               | H-channel L1 Distance | H-channel Correlation | t-LPIPS |
>     |:----------------------|:---------------------:|:----------------------:|:-------:|
>     | w/o Re-encode/Decode  |      0.697            |         0.675          |  0.059  |
>     | MMPL (Full)           |      **0.028**            |         **0.999**          |  **0.023**  |
>
>     These results show that re-encode/decode effectively eliminates cross-segment color drift. (as indicated by the much better H-channel metrics and the nearly unchanged LPIPS score).
>
> 2. **Noise-Interpolation Initialization:**
> This strategy suppresses content jumps at the transitions from planning frames to content frames. We disable it and evaluate boundary quality at these transition points:
>
>     | Variant                 | H-channel L1 Distance | H-channel Correlation | t-LPIPS |
>     |:------------------------|:---------------------:|:----------------------:|:-------:|
>     | w/o Noise-Interpolation |       0.187         |      0.976           |  0.416   |
>     | MMPL (Full)             |       **0.051**         |      **0.998**           |  **0.030**   |
>
>     These results demonstrate that noise interpolation greatly improves the stability of segment transitions and eliminates content jumps (as reflected by nearly identical H-channel metrics and a significantly reduced LPIPS score).

---

### Official Review · Reviewer_k4xv · 2025-10-31

**Soundness:** 3
**Presentation:** 3
**Contribution:** 3
**Rating:** 4
**Confidence:** 3

**Summary:**

The paper explores long-form autoregressive video generation, where accumulated frame-level errors and strict sequential inference cause temporal drift and poor scalability. It introduces Macro-from-Micro Planning (MMPL), a hierarchical “plan-then-populate” scheme that first predicts sparse keyframes for every short segment (Micro) and then chains these plans across segments (Macro), enabling parallel synthesis of intermediate frames while suppressing error propagation. Extensive comparisons on 30-s clips from the VBench-long benchmark and a 29-participant user study show MMPL outperforming open-source baselines (SkyReelsV2, MAGI, CausVid, Self Forcing) in subject consistency, motion smoothness, aesthetic quality, and human preference, while 4-GPU parallel inference cuts wall-clock time to 33 %. The manuscript is clearly structured.

**Strengths:**

- This paper tackles an important and practical problem—generating long, high-quality videos with autoregressive models—whose limitations in temporal drift and sequential bottlenecks are well analyzed.

- The authors propose an elegant two-level planning framework that decouples long-range dependency modeling from dense frame generation, achieving both consistency and parallelism without architectural surgery.

- Experiments are good, combining automatic metrics, human evaluation, ablations, and efficiency analysis on multi-GPU setups, demonstrating superiority.

- The writing is good, figures are informative.

**Weaknesses:**

My major concern about this paper lies in the rather limited performance improvement demonstrated by the proposed technique. In other words, the claimed state-of-the-art results may largely stem from an unfair comparison. Specifically, the method is trained on an exceptionally strong baseline model—Wan-2.1—yet Table 1 does not include any comparison with Wan-2.1 itself. One possible reason for this omission might be that Wan-2.1 cannot generate videos as long as those presented in this work. However, I believe it is essential to conduct detailed ablation studies on short 5-second videos to demonstrate that the improvements reported in Table 1 indeed originate from the proposed MMPL mechanism, rather than simply from extending a 5-second model to 30 seconds. Another observation reinforcing my concern comes from the relatively weak ablation results in Table 2. For instance, even after removing the modules, the model still achieves SOTA performance on motion smoothness and aesthetic quality in Table 1. Meanwhile, for background consistency and imaging quality, the MMPL approach does not achieve SOTA at all. These results make me question the true effectiveness and generalizability of the proposed technique.

**Questions:**

see weakness

---

> ### Author Response · Authors · 2025-11-26
>
> Dear reviewer k4xv, we sincerely thank for your constructive feedback, insightful questions, and valuable suggestions. Below, please find our responses to your concerns.
>
> > **Weakness 1**: Table 1 should include comparisons with Wan-2.1 on short 5-second videos.
>
>
> This concern is primarily based on a **misunderstanding** of the challenges involved in long video generation. There may be an impression that long video generation can be achieved by ``simply extending a 5-second model to 30 seconds``, leading to the assumption that our SOTA performance is mainly inherited from the strong baseline model, Wan-2.1.
> We clarify that Wan-2.1 is incapable of generating long videos, and that our long video generation capability is primarily attributed to MMPL. To further address this misunderstanding, we additionally compare MMPL with Wan on 5-second videos and with SkyReels-V2 on 30-second videos. These results show that MMPL can generate high-quality long videos while preserving its short-video quality, demonstrating that the proposed planning mechanism improves long-video generation without sacrificing short-video performance.
>
>
> **5-second VBench Quality Metrics**
>
> | Model        | Subj.  | Back.  | Mot.  | Aes.  | Img.  |
> |:-------------|:------:|:------:|:-----:|:-----:|:-----:|
> | Wan-2.1      | 0.980  | 0.970  | 0.988 | 0.600 | 0.671 |
> | **MMPL**     | **0.984** | **0.971** | **0.993** | **0.629** | 0.663 |
>
>
> **30-second Long-Video Evaluation**
>
> | Model        | Support long videos | Subj. | Back. | Mot. | Aes.  | Img.  | H-channel L1 | H-channel Corr |
> |:-------------|:------------------------:|:-----:|:-----:|:----:|:------:|:------:|:--------------:|:----------------:|
> | Wan-2.1      |           ×              |   ×   |   ×   |   ×  |   ×    |   ×    |       ×        |        ×         |
> | Skyreels-v2  |           ✓              | 0.956 | 0.966 | 0.991| 0.600  | 0.581  |     0.902      |      0.503       |
> | **MMPL-14B** |           ✓              | **0.980** | **0.968** | **0.992** | **0.628** | **0.661** |   **0.306**      |    **0.927**      |
>
>
> > **Weakness 2**: Limited performance to achieve SOTA, and relatively weak ablation results.
>
> We sincerely thank the reviewer for raising these critical points regarding the ablation study and performance validity. We appreciate the opportunity to clarify and provide further evidence.
>
> 1. Firstly, we acknowledge that the initial ablation analysis may have been insufficient. To further demonstrate the effectiveness of our modules, we conducted additional experiments that isolate the individual effects of each component as well as their combined contributions, making it clearer how and where the performance gains arise.
>
> 2. Secondly, VBench scores alone cannot fully represent a model’s true capability. To address this limitation, we conducted a broader set of experiments that further demonstrates the advantages of our method, particularly in aspects not well captured by VBench.
>
> >> **Weakness 2-1**: Weak ablation results.
>
> Thank you for pointing out the weak ablation results in Table 2. This is because removing certain MMPL components mainly causes occasional local flicker at the planning-frame boundaries. Such localized artifacts do not affect the aggregated VBench scores, since motion smoothness is computed over the entire long video and aesthetic quality is evaluated at the frame level. Therefore, the overall VBench metrics in Table 1 remain high.
>
> To more clearly demonstrate the effectiveness of our modules, we further evaluate temporal color-histogram consistency and t-LPIPS at segment boundaries, following Reviewer b7Dw’s suggestions, which reveals the performance drop when these modules are removed.
>
> 1. For the w/o Early Planning variant, we measure boundary degradation using the discrepancy between the **Initial Frame** and the **Next-Frame** boundary.
>
>     | Variant                 | H-channel L1 Distance | H-channel Correlation | t-LPIPS |
>     |:------------------------|:---------------------:|:----------------------:|:-------:|
>     | w/o Early Planning    |       0.212         |      0.972           |  0.214   |
>     | MMPL (Full)           |       **0.019**         |      **0.999**           |  **0.028**   |
>
>
> 2. For the w/o Mid Planning variant, we measure boundary degradation using the discrepancy between the **Initial Frame** and the **Midpoint** boundary.
>
>     | Variant                 | H-channel L1 Distance | H-channel Correlation | t-LPIPS |
>     |:------------------------|:---------------------:|:----------------------:|:-------:|
>     | w/o Midpoint Planning |       0.545         |      0.889           |  0.347   |
>     | MMPL (Full)           |       **0.064**         |      **0.997**           |  **0.139**   |

---

> > ### Author Response · Authors · 2025-11-26
> >
> > >> **Weakness 2-2**: Limited Metric Improvement Performance with baselines.
> > 1. **Vbench-long Quality Metrics:** We report **VBench-long scores** on the **final 10 seconds of each 30-second** video to assess long-term stability, where maintaining quality is most challenging.
> >
> >     | Model        |  Subj  | Back. |  Mot. |  Aes. | Img. |
> >     |:-------------|:------:|:-----:|:-----:|:-----:|:----:|
> >     | CausVid      | 0.968  | 0.959 | 0.973 | 0.598 | 0.652 |
> >     | Self-Forcing | 0.951  | 0.946 | 0.972 | 0.584 | 0.676 |
> >     | **MMPL**     | **0.972** | **0.961** | **0.989** | **0.613** | 0.651 |
> >
> >     Although VBench provides a comprehensive suite of perceptual metrics, we find that its image-quality score does not always align with human judgments of visual fidelity. To better understand this discrepancy, we conducted a small user study on the same VBench video frames, where participants were asked to choose which individual frames appeared visually superior. The results indicate that models with slightly lower image-quality scores may nonetheless be preferred more frequently by human viewers, while MMPL receives the highest preference despite having an image-quality score comparable to several baselines. This suggests that human-perceived frame quality captures aspects of visual realism that image-quality score alone does not measure, and therefore serves as a useful complementary signal when evaluating generative video quality.
> >
> >     | Model               | Img.  | Quality Preference |
> >     |:--------------------|:-----:|:-----------------------:|
> >     | Self-Forcing (30s)  | 0.689 |          0.21           |
> >     | CausVid (30s)       | 0.661 |          0.10           |
> >     | FramePack-13B (30s) | 0.638 |          0.31           |
> >     | MMPL-14B (30s)      | 0.661 |          0.38           |
> >
> > 2. **Full-Vbench Quality Metrics:** We further report the **standard full VBench benchmark (rather than VBench-long)** on 30-second videos.
> >
> >     | Model                 | Subject Consistency | Background Consistency | Temporal Flickering | Motion Smoothness | Dynamic Degree | Aesthetic Quality | Imaging Quality | Quality Score |
> >     |:----------------------|:-------------------:|:-----------------------:|:--------------------:|:------------------:|:--------------:|:-----------------:|:---------------:|:-------------:|
> >     | CausVid         | 89.50 | 90.00 | 99.41 | 98.06 | 63.88 | 61.82 | 65.30 | 80.89 |
> >     | Self-Forcing    | 88.61 | 89.53 | 98.90 | 98.57 | 68.05 | 60.60 | 68.98 | 81.39 |
> >     | MMPL (ours)     | 92.26 | 94.16 | 99.11 | 98.83 | 61.11 | 62.77 | 65.27 | 82.47     |
> >
> >
> > 3. **Color-shift Metrics:**  Following the suggestion of Reviewer b7Dw, we introduce additional metrics to more comprehensively assess color stability. Specifically, we compute the H-channel L1 distance and H-channel correlation **between the first and last frames** of each 30-second video, providing a direct measure of long-range color drift.
> >
> >     | Model        | H-channel L1 Distance | H-channel Correlation |
> >     |--------------|:----------------------:|:----------------------:|
> >     | CausVid      |          0.711              |          0.598              |
> >     | Self-Forcing |          1.152              |          0.162              |
> >     | FramePack    |          0.445              |          0.881              |
> >     | **MMPL**     |          **0.306**              |          **0.927**              |
> >
> > 4. **Long-Video Consistency Metrics:**  We evaluate JEPA metrics [1] on full 30-second videos to measure temporal coherence over extended durations.
> >
> >     > JEPA is originally designed to estimate dataset-level distributions, and we adopt a simple reinterpretation: treating a single video as a dataset and each frame as a sample. Under this view, taking the first frame as the distributional center, the JEPA standard deviation naturally reflects the amount of temporal drift throughout the video.
> >     (JEPA-Score Std ↓ = more stable; First–Last Frame Diff ↓ = more consistent)
> >
> >     | Model        | JEPA-Score Std | First–Last Frame Score Diff |
> >     |--------------|:---------------:|:----------------:|
> >     | CausVid      |       0.1107          |     0.4093             |
> >     | Self-Forcing |       0.2293          |     1.0695             |
> >     | FramePack    |       0.0853          |     0.1364             |
> >     | **MMPL**     |       **0.0705**          |     **0.0281**             |
> >
> >     [1] Gaussian Embeddings: How JEPAs Secretly Learn Your Data Density, ArXiv 2025.

---

> > > ### Author Response · Authors · 2025-11-26
> > >
> > > 5. **User Study Scores.**  We conduct a user study on 30-second videos to measure human preference on Overall Quality and Long-term Consistency, complementing automated metrics.
> > >
> > >     | Model        | Long-term Consistency | Overall Quality |
> > >     |:-------------|:---------------------:|:---------------:|
> > >     | CausVid      |         0.10              |       0.03          |
> > >     | Self-Forcing |         0.29              |       0.12          |
> > >     | **MMPL**     |         **0.61**          |       **0.85**      |
> > >
> > > 6. **Additional video results.** We provide additional 30-second video examples to illustrate long-video consistency and robustness against temporal drift, which are not fully captured by metrics alone. Anonymous Link: https://anonymous.4open.science/r/Anonymous_MMPL_Webpage_Submission_Number_456/

---

### Official Review · Reviewer_431X · 2025-10-31

**Soundness:** 3
**Presentation:** 3
**Contribution:** 3
**Rating:** 6
**Confidence:** 5

**Summary:**

- This paper proposes a planning-then-populating framework MMPL. MMPL first creates a global video storyline via two stages: Micro Planning, which predicts keyframes within short segments to guide local generation, and Macro Planning, which links these micro plans autoregressively to ensure long-term consistency. Then, Content Populating generates all intermediate frames in parallel across segments, significantly improving efficiency. Experiments show the method achieves both quality and stability in long-video generation.

**Strengths:**

- MMPL proposes a hierarchical autoregressive planning pipeline for long-video generation.
- MMPL could parallelly synthesizes frames for multiple video segments guided by pre-planned keyframes.
- MMPL incorporates a workload scheduling strategy to minimize the overhead of the proposed pipeline.

**Weaknesses:**

- **Decoupled planning and generation pipelines.** MMPL introduces a typical divide-and-conquer strategy to schedule the long-video generation task. I consider a key challenge lies in the fidelity of motion modeling during the planning stage when complete frames are unavailable (i.e., MMPL seems not to be an end-to-end training method). The authors should report a dynamic degree metric and compare both subject and camera motion against standard full or causal attention baselines.
- **Incomplete VBench metrics.** The authors should report all quality, semantic and overall metrics, even though MMPL is a lightweight fine-tuning method built on WanX-2.1 14B. Additionally, I believe the dynamic degree warrants significant comparison, as it typically involves a trade-off between image quality and object motion.
- **Contrained workload scheduling.** Although MMPL employs a divide-and-conquer strategy to distribute computation tasks, it is inherently multi-GPU friendly. However, on a single GPU, the proposed method requires device kernel–based scheduling, which limits its potential for parallel generation.  Consequently, the authors could further optimize MMPL for single-GPU execution.

**Questions:**

- See weaknesses.

---

> ### Author Response · Authors · 2025-11-26
>
> Dear reviewer 431X, we sincerely thank for your constructive feedback, insightful questions, and valuable suggestions. Below, please find our responses to your concerns.
>
> > **Weakness 1**: The challenge of maintaining motion fidelity when MMPL does not seem to be trained end-to-end.
>
> Thank you for pointing out the concern about motion fidelity. We clarify that MMPL uses exactly the same end-to-end training setup as the causal baseline, and the only change is the attention mask. Therefore, MMPL does not introduce extra difficulty in motion modeling.
>
> 1. To quantitatively evaluate motion fidelity in long-video generation, we report **VBench-long dynamic metrics** on 30-second videos generated by **MovieGen prompts**.
>
>     | Model        | Motion Smoothness  | Dynamic Degree  |
>     |:-------------|:--------------------------:|:------------------------:|
>     | CausVid      |           0.981            |          0.567           |
>     | Self-Forcing |           0.980            |          0.624           |
>     | MMPL         |           0.992            |          0.601           |
>
> 2. Furthermore, we would like to clarify that text-to-video models do not take camera trajectories or action sequences as inputs, nor do they provide any ground-truth camera or motion annotations. For this reason, making fair or frame-aligned quantitative comparisons of “subject and camera motion” is not appropriate in this setting. To address your concern, we generate 30-second videos for all methods using the same prompts (including the camera and motion descriptions) and provide qualitative examples here: https://anonymous.4open.science/r/Anonymous_MMPL_Webpage_Submission_Number_456/
>
> > **Weakness 2**: All quality, semantic and overall metrics should be reported.
>
> Following your recommendation, we evaluate the **standard full VBench benchmark (rather than VBench-long)** on 30-second videos, covering all 946 prompts and all 16 metrics.
>
> | Model                 | Subject Consistency | Background Consistency | Temporal Flickering | Motion Smoothness | Dynamic Degree | Aesthetic Quality | Imaging Quality | Quality Score |
> |:----------------------|:-------------------:|:-----------------------:|:--------------------:|:------------------:|:--------------:|:-----------------:|:---------------:|:-------------:|
> | CausVid         | 89.50 | 90.00 | 99.41 | 98.06 | 63.88 | 61.82 | 65.30 | 80.89 |
> | Self-Forcing    | 88.61 | 89.53 | 98.90 | 98.57 | 68.05 | 60.60 | 68.98 | 81.39 |
> | MMPL (ours)     | 92.26 | 94.16 | 99.11 | 98.83 | 61.11 | 62.77 | 65.27 | 82.47     |
>
> | Model                 | Object Class | Multiple Objects | Human Action | Color | Spatial Relationship | Scene | Temporal Style | Appearance Style | Overall Consistency | Semantic Score |
> |:----------------------|:------------:|:----------------:|:------------:|:-----:|:---------------------:|:-----:|:---------------:|:----------------:|:--------------------:|:--------------:|
> | CausVid      | 78.56 | 58.84 | 81.00 | 81.02 | 59.62 | 31.32 | 22.51 | 20.04 | 23.16 | 65.85 |
> | Self-Forcing | 80.06 | 62.88 | 83.00 | 79.80 | 74.76 | 30.59 | 23.78 | 20.41 | 24.80 | 69.17 |
> | MMPL (ours)  | 78.25 | 57.24 | 80.00 | 82.46 | 73.84 | 29.91 | 24.34 | 19.76 | 24.40 | 67.91 |
>
> Our method achieves **the highest overall Quality Score**, highlighting its clear advantage in long-video generation. Meanwhile, MMPL attains competitive Dynamic Degree performance. We view this as a **desirable balance**: the planning mechanism effectively **strengthens long-range stability and perceptual quality, while still preserving sufficient short-term motion variations**, achieving a well-calibrated trade-off between dynamic expressiveness and long-term consistency. We provide additional 30-second video examples to demonstrate the clear advantages of our method in long-video generation, which are not fully captured by metrics alone. Anonymous Link: https://anonymous.4open.science/r/Anonymous_MMPL_Webpage_Submission_Number_456/
>
> > **Weakness 3**: Limits potential for parallel generation on a single GPU.
>
> We appreciate the reviewer’s concern regarding single-GPU parallelism. In practice, the feasibility of parallel generation on a single device is mainly determined by model size rather than by MMPL itself. For the 1.3B backbone, MMPL can already run multi-segment parallelization on a single GPU. For the larger 14B model, however, even short-video generation uses almost all memory on an 80GB H100, leaving no room for segment-level parallel execution. This limitation comes from the model’s parameter count, not from MMPL. When equipped with higher-capacity GPUs such as the NVIDIA H200, the 14B model is able to perform single-GPU parallelization efficiently.

---

### Comment · Area_Chair_aG8q · 2025-11-27

Dear Reviewers,

This is a gentle reminder to please take a moment to review the author rebuttals and check whether your main concerns have been adequately addressed.

If possible, please update your reviews or add a brief clarification on whether the responses resolved your questions or if any issues remain. Your follow-up feedback is important for ensuring a fair and well-informed decision process.

Thank you again for your time and for helping maintain the quality of the ICLR review process.

Best,
AC

---

### Author Response · Authors · 2025-12-01

## Author General Responses

**We would like to sincerely thank all reviewers for your efforts and valuable comments to improve our work! We have uploaded a new version of our main manuscript and supplementary material, revised based on reviewers' valuable and helpful comments. We highlight the revised parts in blue color for better reference.**

We sincerely appreciate the reviewers’ positive assessments across the different aspects of our method.

1. **Reviewers appreciate our significance, novelty and new insights:**

-  The authors propose an elegant two-level planning framework that decouples long-range dependency modeling from dense frame generation, achieving both consistency and parallelism without architectural surgery. *(Reviewer *`k4xv`*)*
-  Making long-video AR generation as plan-then-populate with keyframes per segment, then parallel population conditioned on those anchors, is a clean, modular perspective. *(Reviewer *`b7Dw`*)*
-  While planning and hierarchical generation exist, the specific coupling of joint keyframe prediction from the segment’s first frame plus segment-level AR chaining is crisply articulated and differs from step-jump or packed-context approaches. *(Reviewer  *`b7Dw`*)*

2. **Reviewers appreciate our method effectiveness, scalability and extensive experiments:**

- MMPL incorporates a workload scheduling strategy to minimize the overhead of the proposed pipeline. *(Reviewer *`431X`*)*
- This paper tackles an important and practical problem—generating long, high-quality videos with autoregressive models. *(Reviewer *`k4xv`*)*
- Experiments are good, combining automatic metrics, human evaluation, ablations, and efficiency analysis on multi-GPU setups, demonstrating superiority. *(Reviewer *`k4xv`*)*
- The stability tooling around re-encode/decode across segment boundaries and an appendix noise-initialization scheme for smoother transitions addresses practical artifacts (color shift/flicker) that often undermine long-video demos. *(Reviewer *`b7Dw`*)*
- The proposed method is simple and easy to implement. It’s an independent strategy that can be easily combined with self-forcing and other inference speed-up techniques. *(Reviewer *`R1mP`*)*

3. **Reviewers appreciate our writing clarity and organization:**
- The manuscript is clearly structured. *(Reviewer *`k4xv`*)*
- The writing is good, figures are informative.  *(Reviewer *`k4xv`*)*
- The two-mode scheduler (minimum memory peak vs. maximum throughput) provides an explicit, practical deployment knob that many papers hand-wave. The interplay between when to start the next segment and how to reuse/skip frames is well described. *(Reviewer *`b7Dw`*)*

---

> ### Author Response · Authors · 2025-12-01
>
> ## Summary of Revisions
> **We thank all reviewers for their careful reading and constructive suggestions.**
>
> **We have uploaded a revised version of the manuscript and supplementary material, with all updated content highlighted in blue. In addition, we provide an anonymous demo page (https://mmpl-website.github.io/MMPL-Website/) together with an anonymous video repository (https://anonymous.4open.science/r/Anonymous_MMPL_Webpage_Submission_Number_456/) to offer more comprehensive visual examples.**
>
> **Below we outline how we have addressed the core concerns from each reviewer and strengthened the work accordingly.**
>
> ## **Reviewer-wise Summary of Concerns and Resolutions**
>
> The reviewer *`431X (Rating: 6, Confidence: 5)`* expresses concern about MMPL’s motion fidelity over long sequences and questions whether MMPL is trained end-to-end. We clarify that MMPL is an end-to-end trained method and provide additional quantitative and qualitative examples demonstrating that MMPL maintains strong motion fidelity in long videos, both on VBench-long and full VBench, covering all 946 prompts and 16 metrics, and our method surpasses CausVid in both motion smoothness and motion degree over long sequences.
>
> The reviewer *`k4xv (Rating: 4, Confidence: 3)`* primarily raises concerns regarding the performance of MMPL. This concern is primarily based on a **misunderstanding** of the challenges involved in long video generation. The reviewer mistakenly believes that long video generation can be achieved by
> > “simply extending a 5-second model to 30 seconds”
>
> , and thus assumes that our SOTA performance in long video generation primarily results from the strong baseline model, Wan-2.1. We clarify that Wan-2.1 is incapable of generating long videos, and that our long video generation capability is primarily attributed to MMPL. Our results show that MMPL maintains comparable or superior short video generation performance to Wan-2.1, while also generating stable 30-second videos with minimal visual drift, outperforming the powerful SkyReels-V2.
>
> The reviewer *`b7Dw (Rating: 6, Confidence: 4)`* primarily expresses concerns about the experimental setup and method efficiency analysis. We conduct extensive comparative experiments with CausVid, Self-Forcing, and FramePack using the same experimental setup, demonstrating MMPL’s clear advantages, particularly in temporal consistency and stability in long video generation. We also conduct a thorough analysis of model efficiency, including latency, average GPU utilization, peak VRAM, and time costs for various MMPL variants.
>
> The reviewer *`R1mP (Rating: 4, Confidence: 4)`* primarily expresses concerns about **typographical errors** in the manuscript. We have made thorough revisions to the writing and figures to enhance clarity and readability.

---

> ### Author Response · Authors · 2025-12-03
>
> ## **Detailed Reviewer-wise Summary of Concerns and Resolutions**
> 1. We appreciate the constructive suggestions from Reviewer *`431X`*. Below, we provide our responses to the concerns and questions raised.
> - **Concern about long-range motion fidelity:** We appreciate the reviewer’s thoughtful observation regarding motion fidelity over long sequences. We clarify that MMPL is an end-to-end trained method and provide additional quantitative and qualitative examples demonstrating that MMPL maintains strong motion fidelity in long videos.
>
> - **Concern about incomplete VBench metrics:** Following the reviewer's suggestion, we conduct a comprehensive evaluation on the full VBench benchmark, covering all 946 prompts and 16 metrics. Specially, MMPL achieves the highest overall Quality Score with comparable semantic scores, demonstrating strong effectiveness for long-video generation.
>
> - **Concern about constrained workload scheduling:** We appreciate the reviewer’s attention to the real-world deployment challenges. Here we clarify that the observed constraint is primarily due to the inference-time memory usage of the base model (Wan‑2.1), which already saturates GPU memory when generating even short videos. MMPL itself introduces no additional computational or memory overhead beyond the base model.
>
> 2. We thank Reviewer *`k4xv`* for the helpful feedback. Below, we address the main concerns and issues brought up in the review.
> - **Concern that the improvements in Table 1 may be due to the base model rather than the proposed MMPL mechanism:** This concern is primarily based on a **misunderstanding** of the challenges involved in long video generation. The reviewer mistakenly believes that long video generation could be achieved by ``simply extending a 5-second model to 30 seconds``, and thus assumes that our SOTA performance in long video generation primarily resulted from the strong baseline model, Wan-2.1. We clarify that Wan-2.1 is incapable of generating long videos, and that our long video generation capability is primarily attributed to MMPL. Our results demonstrate that MMPL maintains comparable or superior short video generation performance to Wan-2.1, while also generating stable 30-second videos with minimal visual drift, outperforming the powerful SkyReels-V2.
>
>
> - **Concern about the performance of the ablation experiment:** Following the reviewer’s suggestion, we introduce additional evaluations using color and content consistency metrics to assess the effectiveness of individual modules. The results show that the full MMPL method clearly improves visual quality and smoothness at segment and frame boundaries, demonstrating the contribution of each module.
>
> 3. We are grateful to Reviewer *`b7Dw`* for the valuable comments. Below, we respond to the key concerns highlighted in the review.
> - **Suggestion about the novelty of our method:** Following the reviewer’s suggestion, we compare MMPL with methods like FramePack and CausVid in terms of **approach**, **efficiency**, and **scalability**. This side-by-side comparison highlights the novelty of MMPL.
>
> - **Request for evaluations under the same settings:** Following the reviewer’s suggestion, we compare MMPL with both autoregressive methods and planning-based methods under the same experimental settings. The results show that our method achieves the best performance in 30-second long-video generation, demonstrating the advantage of our approach.
>
> - **Suggestion about adding detailed computational and efficiency analysis:** Following the reviewer’s suggestion, we provide a detailed analysis of MMPL’s parallel generation, including computational cost and inference speed. Specially, We include results in the updated supplementary material. The experiments show that our method is up to 3× faster than both existing autoregressive approaches and planning-based methods, demonstrating the high efficiency of our approach.
>
> - **Suggestion about adding additional ablation experiments:** Following the reviewer’s suggestion, we conduct ablation experiments on the additional strategies described in the supplementary material. We include the results in the updated manuscript. The experiments show that these strategies significantly improve the stability of long-video generation.

---

> > ### Author Response · Authors · 2025-12-03
> >
> > 4. We sincerely thank reviewer *`R1mP`* for constructive feedback, insightful questions, and valuable suggestions. We have addressed and resolved the reviewers’ concerns as follows.
> > - **Concern regarding demo accessibility:** Our demo is hosted on Anonymous GitHub, which experiences a temporary service issue during the review period. The issue is caused by the hosting platform. We resolve it and add an extra anonymous link to ensure stable access.
> >
> > - **Concern about presentation:** Following the reviewer’s suggestions, we revise the writing and figures for improved presentation clarity. All updated content is highlighted in blue for easy reference. We sincerely thank the reviewer for helping us improve the manuscript.
> >
> > - **Request for evaluations under the same backbone:** Following the reviewer’s suggestion, we compare MMPL with existing baselines using identical backbone settings. We include the results in the updated manuscript. The results show that, at both the 1.3B and 14B scales, MMPL achieves the strongest resistance to temporal drift in long-video generation, demonstrating the advantage of our approach.

---

### Meta-Review · Area_Chair_HxFh · 2026-01-05

**Summary:**

The main reviewers' concerns are the performance of MMPL, its novelty, and the poor writing.

AC went through the paper and do see that the writing has been improved.

In terms of novelty, the AC sees some novelty in the micro and macro stages, and it's comparison to framepack-plan was provided in the rebuttal. For the efficiency novelty that the authors claimed, AC thinks that it is quite inadequate, particularly there is no comparison to SOTA AR methods like "Parallelized Autoregressive Visual Generation" in CVPR25, etc.

This is a good segway to the major concern in AC's opinions. Performance. Even after rebuttal, AC is seeing performance that are not across the board better. For efficiency claim, it seems the authors only compared to its own version without parallelization, and comparisons to say the CVPR25 work AC cited above are not provided.

Overall, AC thinks that this paper remains borderline and is going to recommend reject to uphold ICLR's standards.

**Reviewer Concerns:**

Novelty wise, AC thinks it is somewhat addressed, as well as the overall presentation of the paper.

**Reviewer Scores:**

Reviewers questioning novelty and writing may keep their scores due to the rebuttal.

---

### Decision · Program_Chairs · 2026-01-26

Reject